# Large electronegativity differences between adjacent atomic sites activate and stabilize $ZnIn_2S_4$ for efficient photocatalytic overall water splitting

Xu Xin[1,2,10], Yuke Li [3,10], Youzi Zhang [1,2], Yijin Wang[1,2], Xiao Chi [4], Yanping Wei [5], Caozheng Diao [6], Jie Su [7], Ruiling Wang[1,2], Peng Guo[1,2], Jiakang Yu[1], Jia Zhang [3], Ana Jorge Sobrido [8], Maria-Magdalena Titirici [9] & Xuanhua Li [1,2] ✉

Photocatalytic overall water splitting into hydrogen and oxygen is desirable for long-term renewable, sustainable and clean fuel production on earth. Metal sulfides are considered as ideal hydrogen-evolved photocatalysts, but their component homogeneity and typical sulfur instability cause an inert oxygen production, which remains a huge obstacle to overall water-splitting. Here, a distortion-evoked cation-site oxygen doping of $ZnIn_2S_4$ (D-O-ZIS) creates significant electronegativity differences between adjacent atomic sites, with $S_1$ sites being electron-rich and $S_2$ sites being electron-deficient in the local structure of $S_1$–$S_2$–O sites. The strong charge redistribution character activates stable oxygen reactions at $S_2$ sites and avoids the common issue of sulfur instability in metal sulfide photocatalysis, while $S_1$ sites favor the adsorption/desorption of hydrogen. Consequently, an overall water-splitting reaction has been realized in D-O-ZIS with a remarkable solar-to-hydrogen conversion efficiency of 0.57%, accompanying a ~ 91% retention rate after 120 h photocatalytic test. In this work, we inspire an universal design from electronegativity differences perspective to activate and stabilize metal sulfide photocatalysts for efficient overall water-splitting.

The utilization of sunlight and water, two of the most abundant natural resources on earth, for the production of hydrogen ($H_2$) and oxygen ($O_2$) at a stoichiometric ratio of 2:1, holds great potential for achieving carbon neutrality[1]. Compared with the certain solar hydrogen production techniques, such as photo-electrochemical water splitting, photocatalytic overall water-splitting eliminates the need for external bias or circuitry, thereby reducing system costs and mitigating photocatalyst corrosion, stability, and safety concerns[2]. Semiconductor-based photocatalytic overall water splitting is an ideal solar-to-chemical energy conversion route[3]. Constructing a hybrid photocatalyst can enhance light harvesting and facilitate charge separation[4–6]. However, the long reaction paths and random distribution of active sites in hybrid systems limited its photocatalytic activity[7–12]. Recently, single photocatalysts, such as $SrTiO_3$, GaN, $Y_2Ti_2O_5S_2$, $F-C_3N_4$, TpBpy covalent organic frameworks, $SrTaO_2N$, and $Ta_3N_5$, etc. have been developed to achieve overall water splitting while avoiding the problems of constructing hybrid systems[13–20]. However, the development of photocatalysts with high solar-to hydrogen (STH) efficiencies in solar hydrogen systems remains a fundamental challenge[21–23].

Metal sulfides are considered as promising photocatalysts due to their appropriate energy bands, designable structures, and excellent photoelectric properties[24–26]. The representative metal sulfides such as $ZnIn_2S_4$, $MoS_2$, $WS_2$, and $In_2S_3$ have been widely used in the field of photocatalytic water splitting[27–29]. Among them, $ZnIn_2S_4$ (ZIS) is a typical ternary layered metal chalcogenide semiconductor, possessing suitable band gap of about 2.44 eV and conduction band potential of −0.43 eV versus Normal Hydrogen Electrode (NHE), which holds the visible-light absorption and strong reduction capacity for $H_2$ generation from water splitting[24]. Many strategies have achieved enhanced $H_2$ production performance around ZIS photocatalyst, such as constructing a Z-scheme heterostructure of sulfur vacancies ZIS with other semiconductor[9], fabricating S vacancy induced with atomic Cu doping in ZIS nanosheets[23], incorporating anion-site oxygen doping into the sulfur atom sites of ZIS[24], and modulating cocatalyst of protruding single Pt atoms[7], but they have not been able to attain efficient and stable overall water-splitting reactions. One of the critical challenges faced in enhancing the efficiency for water splitting is the homogeneous composition of active sites and consistent electronic structure in single photocatalyst. Additionally, the sulfur atoms present in a ZIS photocatalyst are highly prone to oxidation by photogenerated holes, leading to its instability. Those bottlenecks result in an inert base that hinders oxygen production and ultimately lead to poor overall water-splitting performance[21,22].

Here, a distortion-evoked cation-site O doping of ZIS (D-O-ZIS) was designed to break the homogeneity of its component between adjacent atomic sites and realize high performance overall water splitting. Normally, O atoms tend to occupy the anion-site position of ZIS, where no oxygen is produced (Supplementary Fig. 1). O doping in the cation-site position is challenging to achieve due to the unfavorable energetics involved. Different from the typical anion-site O doping in ZIS, D-O-ZIS overcomes the high barrier of cation-site O doping by constructing an intermediate, distorted high-energy structure, which facilitates the doping of O atoms into the cation sites (i.e., distortion-evoked cation-site O doping). The strategy involves thermally inducing atomic migration to generate a distorted edge structure (D-ZIS), which is subsequently treated with $O_2$ plasma to evoke cation-site O doping and create D-O-ZIS (see Fig. 1a and the methods section for a detailed synthesis process). The distortion states and cation-site O doping induced charge redistribution and altered the electronegativity balance of coordinated atomic sites in the O-doped distortion regions. Specifically, the electron-rich $S_1$ sites and the electron-deficient $S_2$ sites in the local $S_1$-$S_2$-O configuration of D-O-ZIS have manifested optimal adsorption/desorption behavior of $H_2$ or $O_2$ during the reaction. The S-O bond of D-O-ZIS, being a hybridized electronic state of S $3p$-O $2p$, promotes stable oxygen evolution for overall water-splitting reactions and avoids the common issue of sulfur instability in metal sulfide photocatalysis. Consequently, D-O-ZIS as a single photocatalyst exhibited outstanding photocatalytic overall water-splitting performance with a 0.57% STH, accompanied by an enhanced water-splitting stability of ~91% retention rate after 120 h. In this work, this strategy could also effectively activate the oxygen inert of other metal sulfide photocatalysts, such as $MoS_2$ and $In_2S_3$, demonstrating the design universality of the metal sulfides from electronegativity differences perspective for overall water splitting.

## Results

### Photocatalyst characterization

The photocatalysts were analyzed using scanning electron microscopy (SEM) and transmission electron microscopy (TEM), which revealed nanoflower-like structures consisting of nanosheets (Fig. 1b, c, Supplementary Fig. 2). High-resolution TEM (HRTEM) images confirmed that the lattice fringes of the sample had an interplanar distance of 0.32 nm, corresponding to the (102) lattice plane of hexagonal phase ZIS (Fig. 1d–f)[8]. D-ZIS and D-O-ZIS exhibited distorted edge shells

(Fig. 1d, e, yellow squares) compared to ZIS (Fig. 1f). Line profiles indicated that the thicknesses of D-ZIS and D-O-ZIS distorted edge shells were $1.3 \pm 0.3$ and $2.2 \pm 0.2$ nm, respectively (Fig. 1g). Energy-dispersive X-ray spectroscopy (EDX) revealed that Zn, In, S, and O were uniformly dispersed spatially in D-O-ZIS (Fig. 1h), while O atom was hardly detected in ZIS and D-ZIS (Supplementary Fig. 2e, g). Selected area electron diffraction (SAED) patterns further indicated that the structures of D-O-ZIS and D-ZIS were distorted at the edges compared to the ZIS crystal structure (Fig. 1i, Supplementary Fig. 2f, h). High-angle annular dark field scanning TEM (HAADF-STEM) line scans for the Zn element showed that Zn vacancies were confined to the edges of D-ZIS and D-O-ZIS, whereas the line scan for O showed that the O atoms doped in D-O-ZIS were primarily localized at the outer edge (Fig. 1j, Supplementary Fig. 2i), and the crystal structure of the D-O-ZIS is consistent with that of ZIS (Supplementary Fig. 2j).

We conducted a more detailed analysis of the edge shell structure and O doping characteristics of the photocatalysts. The distorted edge shell structure of D-ZIS, which differed from the ZIS without distortion, was primarily caused by S-S bonds and the introduction of Zn vacancies. S $2p$ X-ray photoelectron spectroscopy (XPS) exhibited higher energy shifts in D-ZIS, with additional peaks assigned to the S-S bond ($S_2^{2-}$) at 164.4 and 165.3 eV (Fig. 2a)[26]. Electron spin resonance (ESR) spectroscopy of D-ZIS displayed a peak intensity at $g = 2.004$ attributed to the unpaired free electrons trapped in Zn vacancy, confirming the existence of Zn vacancy (Fig. 2b), with a concentration of ~2.3% determined by Zn XPS spectra (Supplementary Fig. 3, Supplementary Table 1)[26], which is consistent with the inductively coupled plasma (ICP) emission spectrometer results (Supplementary Table 2). The Fourier-transform of $k^3\chi(k)$ curves for Zn $K$-edge extended X-ray absorption fine structure (FT-EXAFS) spectra showed a Zn-S peak at $R = 1.90$ Å in D-ZIS, but with reduced intensity compared to ZIS (Fig. 2c), indicating the presence of Zn vacancy induced distortion states[23]. This finding was further confirmed by the Zn $K$-edge EXAFS spectra in $k$ space, which exhibited a damped oscillation for D-ZIS compared to ZIS (Fig. 2d), and the strong wavelet contour plots (Supplementary Fig. 4)[23,30–33]. The distorted lattice parameters of the Zn $K$-edge EXAFS fitted results (Supplementary Fig. 5, Supplementary Table 3) confirmed the presence of distortion states induced by the S-S bonds and the altered bond lengths[34–36].

The incorporation of O atoms induced the formation of more Zn vacancies, resulting in increased distortion states in D-O-ZIS. XPS and Raman spectroscopy confirmed O doping in D-O-ZIS, with an O atom concentration of 5.8% as confirmed by O $1s$ spectra (Fig. 2a, Supplementary Fig. 3). ESR and Zn $K$-edge FT-EXAFS data suggested an increase in Zn vacancies in D-O-ZIS (Fig. 2b, c), with a concentration of ~3.7% (Supplementary Fig. 3c). A weak peak at about 1.39 Å assigned to the Zn-O coordination in D-O-ZIS[24]. The Zn $K$-edge EXAFS of D-O-ZIS in $k$ space showed minimal oscillation, indicating an increase in distortion states in the structure (Fig. 2d)[34,37]. This is consistent with the formation of Zn vacancies induced by O doping in theoretical calculations (Supplementary Fig. 6). We further verified cation-site O atom doping in Zn atom sites of D-O-ZIS using S $K$-edge X-ray absorption near-edge structure (XANES) spectra, which revealed an S-O coordinated bond at 2481.8 eV (Fig. 2e)[34,36]. The rising S pre-edge showed a higher energy shift from 2467.5 for pristine ZIS to 2468.9 eV for D-ZIS due to S-S anti-bond formation. Doping with O, which possesses stronger electronegativity, led to a further shift of the S pre-edge to 2470.5 eV for D-O-ZIS, generating a higher valence state of coordinated S atom[34], and hence, the electronegativity difference within the S-O bond. The O $K$-edge XANES spectra verified the existence of the S-O bond as an S $1s$–O $2p$ $\delta^*$ anti-bond (Fig. 2f)[34,37]. Density Functional Theory (DFT) calculation simulations of distortion states and O doping structures (Fig. 2g) revealed that the lattice parameters matched the XANES fitted results (Supplementary Fig. 5, Supplementary Table 3). Distortion-evoked cation-site O doping in ZIS has a formation energy ($E_f$) of 5.99 eV for

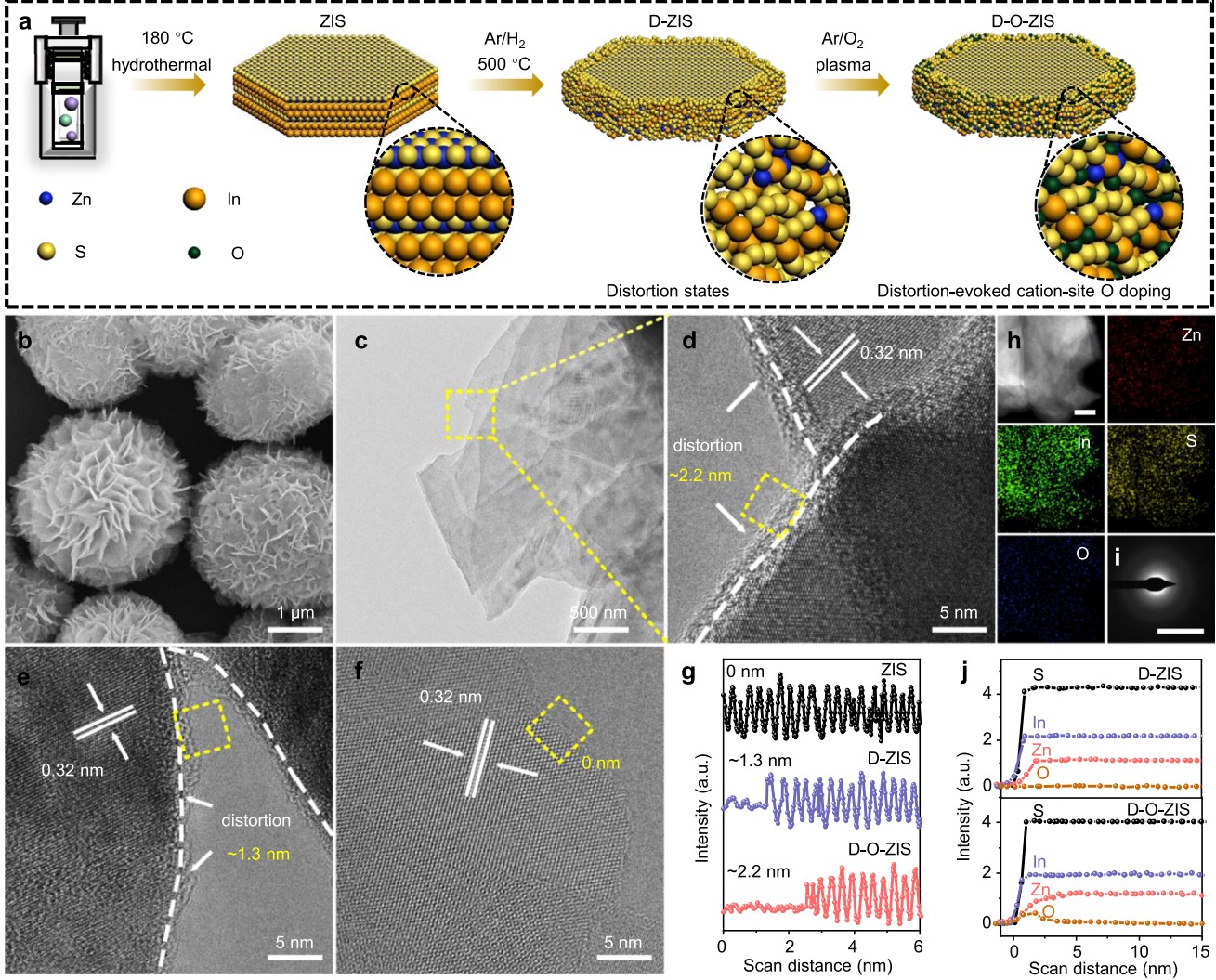

**Fig. 1 | Photocatalyst synthesis and morphology characterization. a** Schematic of the synthetic process for ZIS, D-ZIS, and D-O-ZIS. The yellow, blue, orange, and green spheres represent the S, Zn, In, and O atoms, respectively. The enlarged image shows the edge structure of samples; **b** SEM image of D-O-ZIS; **c** TEM image of D-O-ZIS; **d** HRTEM image of D-O-ZIS (The lattice fringe enlargement of Fig. 1c). The arrow denotes the distortion in edge. The yellow dashed square denotes shell thickness of ~2.2 nm for D-O-ZIS; **e** HRTEM image of D-ZIS. The arrow denotes the distortion in edge. The yellow dashed square denotes shell thickness of ~1.3 nm for D-ZIS; **f** HRTEM image of ZIS. The yellow dashed square denotes shell thickness of ~0 nm for ZIS; **g** The respective line profiles on the edge of ZIS, D-ZIS, and D-O-ZIS from the outer edge to the core in Fig. 1d–f; **h** The corresponding EDX mapping images of D-O-ZIS. The scale bar is 500 nm; **i** SAED pattern of D-O-ZIS edge. The scale bar is 5 nm⁻¹; **j** The element distribution and HAADF-STEM line scans of Zn, In, S, and O elements from the outer edge to the core for D-ZIS and D-O-ZIS. Source data are provided as a Source Data file.

distorted configurations with high-energy structures containing S-S bonds and Zn vacancies. Cation-site O doping has a negative energy of −3.45 eV. This suggests that distortion structures lower the energy required for cation-site O doping in Zn atom sites, making it energetically favorable.

**Photocatalytic overall water-splitting performance**
The photocatalytic performance of samples was evaluated for an overall water-splitting reaction from pure water with Pt and $CoO_x$ used as co-catalysts. Loading of co-catalysts can greatly enhance the photocatalytic activity of D-O-ZIS for water splitting by constructing matched energy band between D-O-ZIS and cocatalysts and reducing the free energy of hydrogen and oxygen adsorption (Supplementary Figs. 7–9). When exposed to light irradiation at AM1.5 G (100 mW cm⁻²) with optimized Pt and $CoO_x$ loading, $H_2$ and $O_2$ were steadily produced over D-O-ZIS, achieving evolution amounts up to 373.2 and 177.6 µmol within 12 h reaction, respectively (Supplementary Figs. 9, 10, Fig. 3a). Meanwhile, during the photocatalytic process, $H_2$ and $O_2$ are less

prone to adsorb onto D-O-ZIS/Pt/$CoO_x$, accompanied by a 14% reduction observed after a 12-h dark reaction (Supplementary Figs. 11–13)[1,38,39]. ZIS and D-ZIS cannot split water into $H_2$ and $O_2$ with co-catalyst loading and activity attenuation occurs in each cycle (Supplementary Fig. 14). The apparent quantum yield (AQY) of D-O-ZIS for overall water splitting was investigated (Supplementary Fig. 15, Supplementary Table 4), and calculated to be 14.90% at 400 nm (Fig. 3b, Supplementary Table 5), higher than that of ZIS (1.40%) or D-ZIS (3.11%) (Supplementary Fig. 16, Supplementary Tables 6, 7). The solar-to hydrogen (STH) efficiency was measured at AM1.5 G (100 mW cm⁻²) simulated sunlight irradiation with a mean value of 0.57% (Fig. 3c, Supplementary Table 8), which outperforms most of the recently reported single photocatalysts (Supplementary Fig. 17, Supplementary Table 9) and ZIS based composite photocatalysts (Supplementary Fig. 18, Supplementary Table 10).

We further investigated the photocatalytic performance of single D-O-ZIS without adding Pt and $CoO_x$ cocatalysts (Supplementary Fig. 19, Supplementary Table 11). The system still

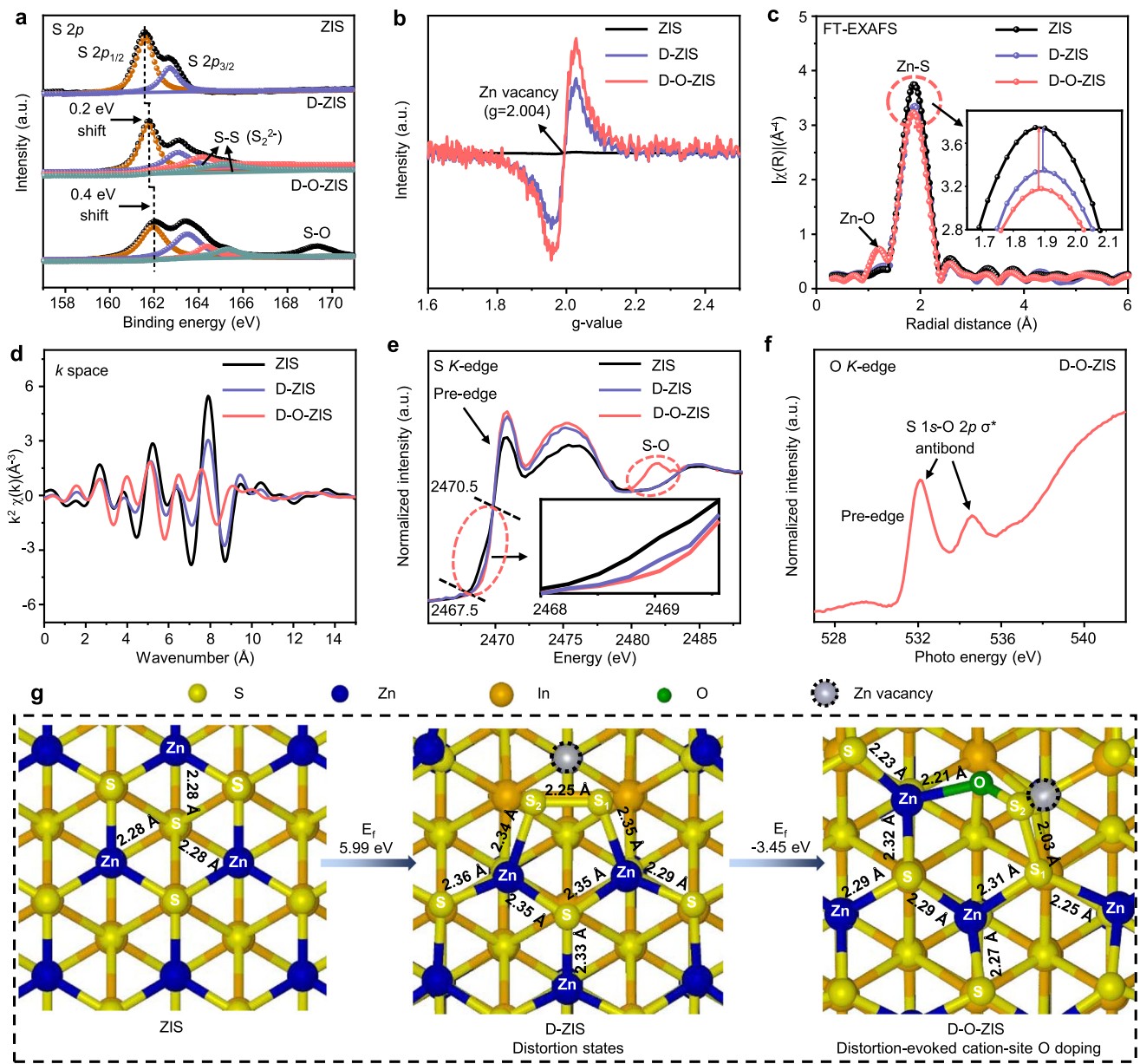

**Fig. 2 | Geometric and local electronic structures of photocatalysts. a** XPS spectra of S 2*p* in ZIS, D-ZIS, and D-O-ZIS; **b** ESR spectra of ZIS, D-ZIS, and D-O-ZIS; **c** The Fourier-transform curves of $k^3$-weighted Zn *K*-edge EXAFS spectra of ZIS, D-ZIS, and D-O-ZIS; **d** Zn *K*-edge EXAFS spectra of ZIS, D-ZIS, and D-O-ZIS in *k* space; **e** Normalized S *K*-edge XANES spectra of ZIS, D-ZIS, and D-O-ZIS. Embedded is an enlargement of S *K*-edge pre-edge; **f** Normalized O *K*-edge XANES spectra of D-O-ZIS; **g** The schematic process of the local structure transformation of D-O-ZIS to form distortion and cation-site O doping; Meanwhile, the corresponding bond lengths were depicted on the structures. The structures are shown in top view. Source data are provided as a Source Data file.

produced $H_2$ and $O_2$ evolution, with values of 76.8 and 36.0 µmol, respectively. The STH efficiency yielded value of 0.12% (Supplementary Table 12) and is the highest of the investigated photocatalysts without loading any cocatalysts (Supplementary Fig. 17). In contrast, ZIS and D-ZIS only produced $H_2$ and showed a decrease in catalytic activity in each cycle (Supplementary Fig. 20). The overall water-splitting activity of D-O-ZIS was further confirmed by performing $H_2$ or $O_2$ evolution half-reactions. D-O-ZIS exhibited $H_2$ or $O_2$ evolution during the half-reactions, while no $O_2$ was detected on ZIS and D-ZIS (Supplementary Fig. 21). The $^{18}O$ isotopic measurement for D-O-ZIS confirmed that the generated $O_2$ was due to water splitting (Supplementary Fig. 22)[4].

We also evaluated the photocatalytic stability test for the ZIS, D-ZIS, and D-O-ZIS after a 120 h reaction. The results showed that D-O-ZIS retained ~91% of its original photocatalytic gas evolution rate,

demonstrating stability of overall water-splitting performance (Fig. 3d), while ZIS and D-ZIS decay almost to zero. The photocorrosion degree of photocatalysts induced by S leaching after the photocatalytic reaction was evaluated by S 2*p* XPS (Fig. 3e). The S 2*p* spectra of D-O-ZIS showed the smallest binding energy shift of 0.02 eV while preserving the pretest intensity compared to ZIS (0.62 eV shift) and D-ZIS (0.59 eV shift) with reduced intensities and oxidation product of $SO_4^{2-}$, indicating that S leaching in D-O-ZIS was significantly suppressed[26,29]. The HRTEM image and structural analysis of D-O-ZIS after testing revealed stable distortion features (Fig. 3f, Supplementary Fig. 23), maintaining the shell thickness of about 2 ± 0.2 nm, while ZIS and D-ZIS exhibited S leaching characteristics (Supplementary Fig. 24). Additionally, we found that S *K*-edge and O *K*-edge XANES for D-O-ZIS showed negligible changes in characteristic peaks after testing (Supplementary Fig. 25), further indicating its structural stability for

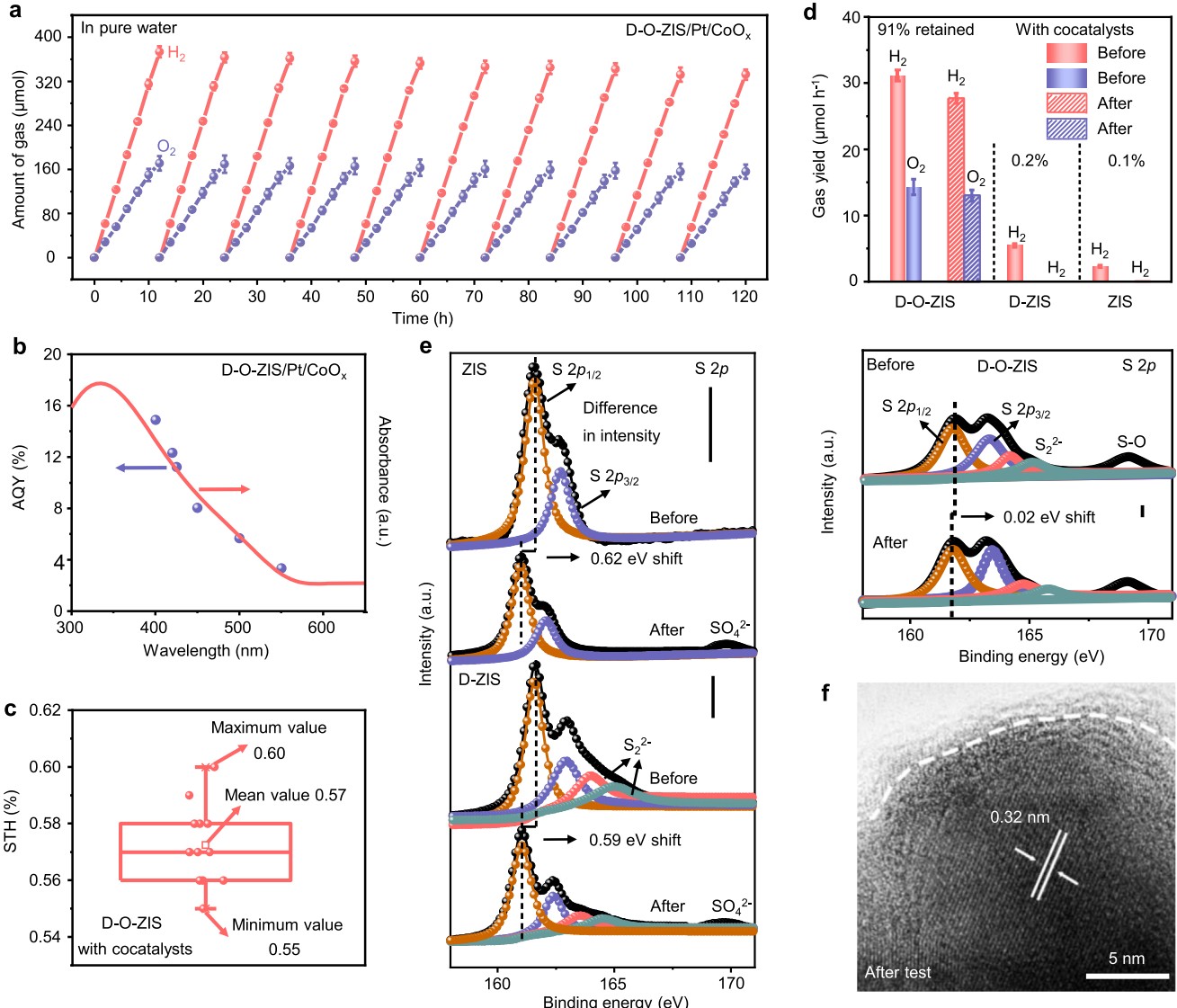

**Fig. 3 | Photocatalytic overall water-splitting performance. a** Time-dependent photocatalytic overall water splitting over D-O-ZIS in pure water under standard AM1.5 illumination (100 mW cm⁻²), Pt and CoOₓ used as cocatalysts, Pt to CoOₓ wt% ratio of 1:4, the photocatalyst mass was 35 mg and the photocatalytic activity was evaluated via the total hydrogen and oxygen yield of a cycle, the time of each cycle is 12 h. Error bars represent the standard deviations from the statistic results of three sets of experiments; **b** Wavelength-dependent of AQY during photocatalytic overall water splitting based on D-O-ZIS. AQY denotes the apparent quantum yield that was calculated using equations (2) and (3) in Supplementary Information and details shown in Supplementary Table 5, Pt and CoOₓ used as cocatalysts, Pt to CoOₓ wt% ratio of 1:4; **c** The STH efficiency of D-O-ZIS with cocatalysts (Pt, CoOₓ) loading for photocatalytic overall water splitting. The STH value was evaluated 12

times with separate samples as shown in Supplementary Table 8 and calculated using Eq. (1) in main text. The center line represents the median, the top and bottom box represent the upper and lower quartile, respectively, the small rectangle represents the mean value and the maximum/minimum values are indicated by the top/bottom bars; **d** Photocatalytic gas yield of ZIS, D-ZIS, and D-O-ZIS before and after photocatalytic overall water splitting test in pure water. Error bars represent the standard deviations from the statistic results of three sets of experiments; **e** The S 2*p* XPS spectra of ZIS, D-ZIS, and D-O-ZIS before and after 120 h photocatalytic test. The vertical bars indicate the difference in intensity before and after test; **f** HRTEM image of D-O-ZIS after 120 h photocatalytic test. Source data are provided as a Source Data file.

photocatalytic reactions (further discussion of stability mechanism in photocatalytic working principle of activation and stability section). This design strategy can activate the oxygen-inert properties of other metal sulfide photocatalysts, including MoS₂ and In₂S₃. This demonstrates the universal applicability of metal sulfides in overall water splitting from an electronegativity difference perspective (Supplementary Fig. 26).

**Kinetics of charge transport and separation**
The optical and electrical characteristics of the samples were investigated to determine the mechanism of the improved photocatalytic activity. The UV–vis absorption spectrum of D-O-ZIS showed intense

optical absorption of the visible region and a redshift compared to ZIS (Supplementary Fig. 27). By introducing distortion and O doping into ZIS, D-O-ZIS exhibited the smallest carrier transport activation energy of 0.13 eV (Fig. 4a), which is favorable for charge transport and was calculated using the Arrhenius equation (Supplementary Fig. 28)[40,41]. To evaluate the carrier separation dynamics, the internal electric field intensity of photocatalysts was analyzed via the potential shift from 0 V to the bias intersection voltage ($V_i$) based on the conductivity test[42–45]. D-O-ZIS showed the strongest internal electric field with the largest potential shift at −0.20 V (Fig. 4b, Supplementary Fig. 29). The internal electric field intensity was further estimated using transient photoelectric measurements and the intensity of D-O-ZIS was 5.1 and

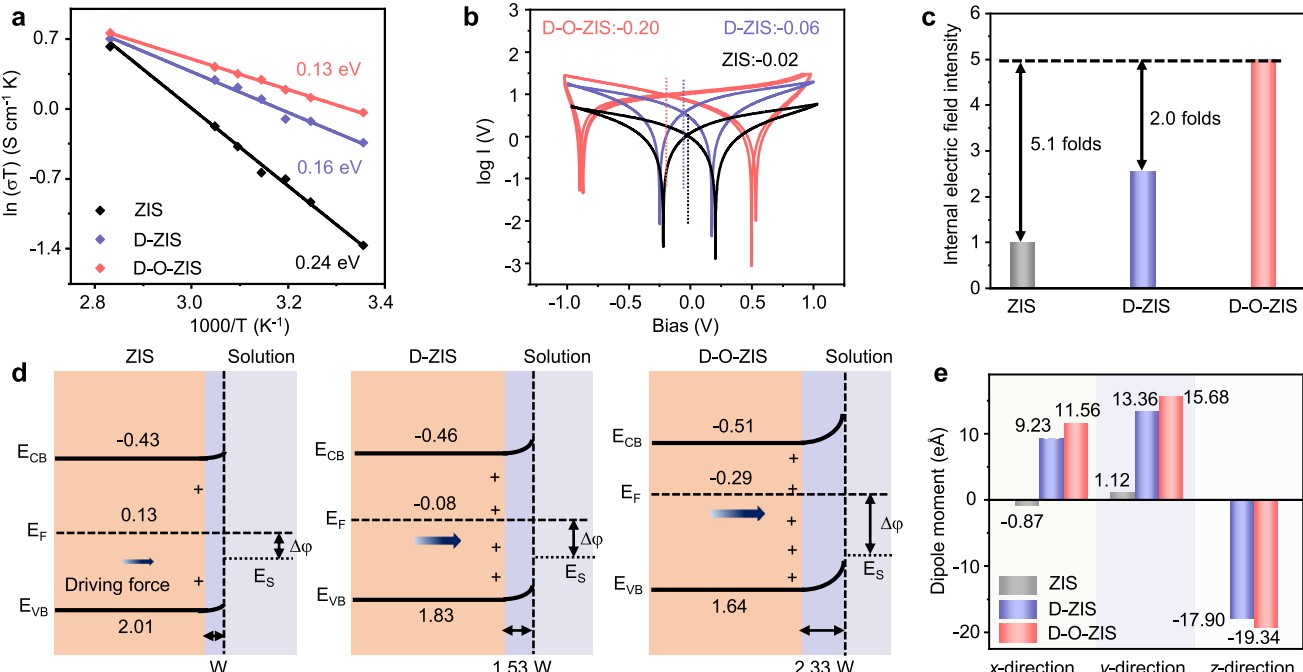

**Fig. 4 | Charge transport and separation kinetics. a** Carrier transport activation energy of ZIS, D-ZIS, and D-O-ZIS (derived from in-situ Electrochemical impedance spectroscopy plots in Supplementary Fig. 28); **b** Electronic conductivity measured via cyclic voltammetry at scanning rate 50 mV s⁻¹. The black dotted line represents the $V_i$ of ZIS. The blue dotted line represents the $V_i$ of D-ZIS, and the red dotted line represents the $V_i$ of D-O-ZIS. The plots are derived from Supplementary Fig. 29; **c** Internal electric field intensity of ZIS, D-ZIS, and D-O-ZIS (according to the Supplementary Fig. 30, assuming the intensity of ZIS to be "1"); **d** Schematic of the detailed band structures, band bending, and space charge region for ZIS, D-ZIS, and D-O-ZIS. Energy of conduction band: ($E_{CB}$), valence band ($E_{VB}$) and Fermi level ($E_F$)

are depicted in diagram, potential versus Normal Hydrogen Electrode (NHE). A minimal degree of band bending and driving force exist on the surface of ZIS due to the dangling bond with unsaturated sulfur atoms on its surface. A mild band bending can be detected in D-ZIS. The D-O-ZIS has significantly strong band bending. $E_F$ difference between the photocatalyst and solution: $\Delta\varphi$, Fermi level of water solution ($E_S$). The detailed energy band calculation of photocatalyst seen in Supplementary Fig. 7. The width of space charge region calculation is obtained from Supplementary Fig. 7b and details seen in Supplementary Information. **e** The calculated dipole moments of ZIS, D-ZIS, and D-O-ZIS along three different structural directions. Source data are provided as a Source Data file.

2.0 times stronger than that of ZIS and D-ZIS, respectively (Fig. 4c, Supplementary Fig. 30)[42]. Additionally, D-O-ZIS exhibited the greatest charge separation ability with a prolonged average carrier lifetime of 42.71 ns and charge separation efficiency of 39.6%, influenced by the internal electric field (Supplementary Fig. 31)[42].

We calculated the energy band during the photocatalytic process to elucidate the enhanced internal electric field intensity and kinetics of charge transport and separation. The energy band structures were determined through UV−vis diffuse reflectance spectra and ultraviolet photoelectron spectroscopy (UPS) (Fig. 4d)[43]. The Fermi level ($E_F$) of the photocatalyst was found to upshift with increasing distortion states and O doping. During the photocatalytic process, the photocatalyst and solution with different electric potentials are in contact ($E_F$: photocatalyst; $E_S$: 0.34 eV for water solution), which creates a space-charge region at the interfaces[46,47]. We estimated the space charge region width ($W$) using a Mott–Schottky plot[46] from Supplementary Fig. 7b and found that D-ZIS and D-O-ZIS showed wider widths of 1.53 and 2.33 $W$ when the depletion region width of ZIS was set as $W$. The enhanced space charge region width is ascribed to the increased $E_F$ difference ($\Delta\varphi$), which is consistent with the trends of the internal electric field. The wider space-charge regions between the photocatalyst and solution interfaces reduced hole drift distance on D-O-ZIS, which provided a strong driving force for charge separation[40,42].

The changes in average potentials and internal electric field intensity were further verified by DFT. The $E_F$ raised in D-O-ZIS due to the increased distortion states and O doping (Supplementary Fig. 32), which is consistent with experimental findings. The electrostatic potential difference reflected the internal electric field intensity, which increased proportionally to the $E_F$ difference between the

photocatalyst and solution interfaces (Supplementary Fig. 33). Further analysis of the dipole moment revealed that the dipole moment changed significantly along the x, y, and z directions in D-O-ZIS, inducing a dipole of structure and adding asymmetry to the local structure, thus increasing the internal electric field (Fig. 4e)[40]. These findings confirmed that the increasing distortion states and O doping in D-O-ZIS enhanced the internal electric field, providing a strong driving force for charge separation.

**Photocatalytic working principle of activation and stability**

In-situ Raman spectroscopy was used to monitor the changes occurring on the surface of D-O-ZIS. The Raman signal for S-H adsorption at 2519 cm⁻¹ was observed over the potential range from 0.02 to −0.12 V versus Reversible Hydrogen Electrode (RHE) to track hydrogen evolution process (Fig. 5a)[48–50]. The S-H peak appeared at 0.02 V and became stronger as the potential increased. The peak at 1194 cm⁻¹ corresponding to *OOH adsorption on D-O-ZIS during oxygen evolution was observed in Fig. 5b[48], which intensified gradually over the anodic potential range from 0.06 to 0.25 V versus RHE. The redshifts of S-H (from 2519–2535 cm⁻¹) and *OOH (from 1194–1210 cm⁻¹) were attributed to the significant stark tuning phenomenon and adsorption of interfacial species, including $H_2O$, $OH^-$, and $H^+$ species[50]. Two-dimensional contour plots also showed that the S-H and *OOH vibrations on D-O-ZIS intensified with time (Fig. 5c, d). The determined shift rates for the S-H and *OOH vibrations in D-O-ZIS are 114 cm⁻¹ V⁻¹ and 84 cm⁻¹ V⁻¹, respectively (Fig. 5e). Raman signals (S-H and *OOH) were detected at −0.12 V and 0.25 V potentials, respectively, for ZIS, D-ZIS and D-O-ZIS (Fig. 5f). The S-H signal was stronger for D-O-ZIS than that of ZIS or D-ZIS, and the *OOH signal was only detected for D-O-ZIS.

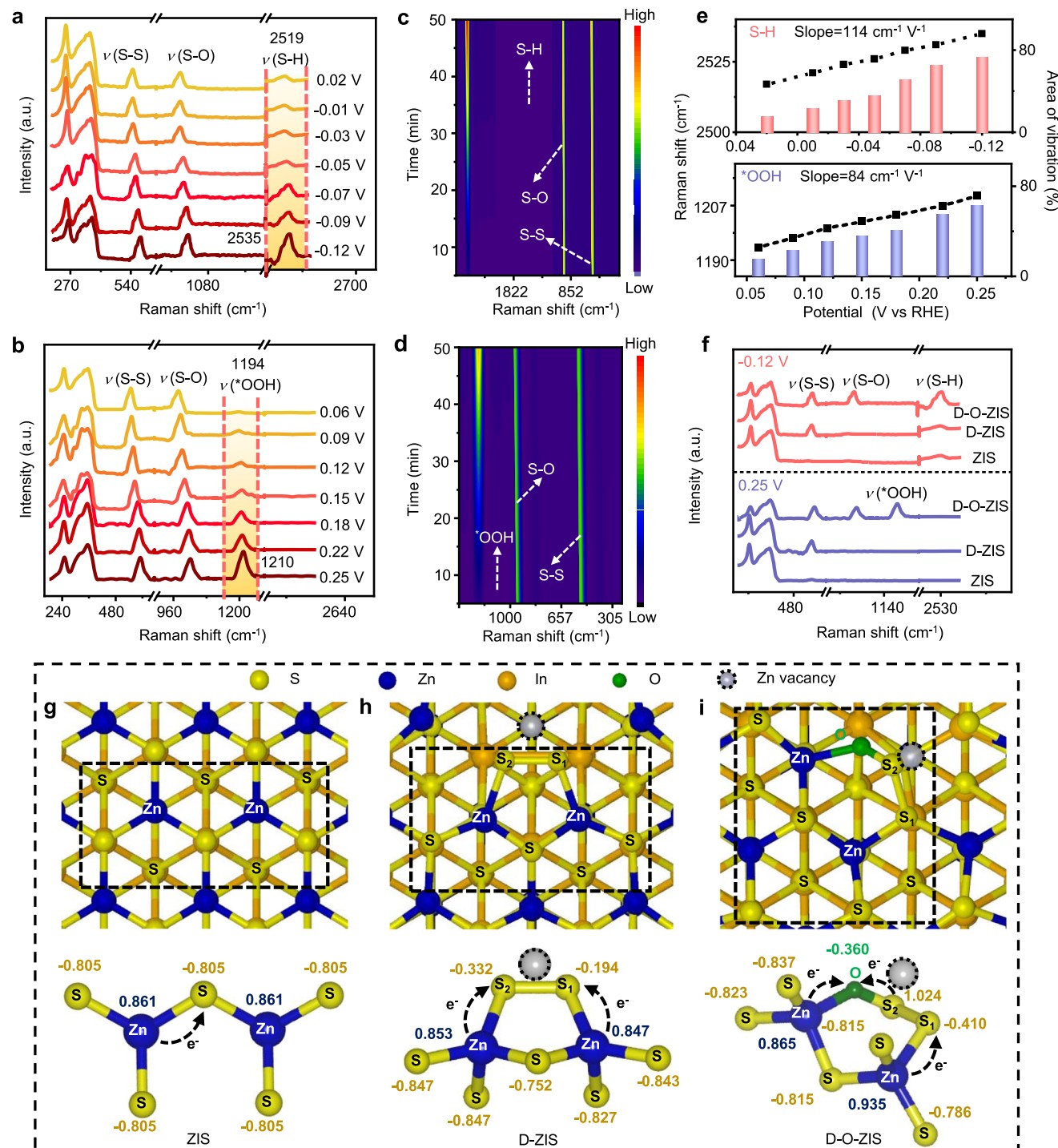

**Fig. 5 | Photocatalytic hydrogen and oxygen evolution process investigated by in-situ Raman spectra. a** In-situ Raman spectra of photocatalytic hydrogen evolution process on D-O-ZIS. A series of Raman spectra at different potential (0.02−−0.12 V versus RHE) exhibit the dynamic variation of hydrogen evolution process; **b** In-situ Raman spectra of photocatalytic oxygen evolution process on D-O-ZIS. A series of Raman spectra at different potentials (0.06−0.25 V versus RHE) exhibit the dynamic variation of oxygen evolution process; **c** 2D contour maps of Raman vibrations of hydrogen evolution process; **d** 2D contour maps of Raman vibrations of oxygen evolution process; **e** Raman shifts and area ratios of the S-H bonds and *OOH vibrations versus potentials at the D-O-ZIS surface; **f** Intensity difference of the Raman signals of S-H vibrations and *OOH vibrations versus identical potentials of −0.12 V and 0.25 V, respectively, for the samples during hydrogen evolution and oxygen evolution processes. The charge on atoms from Bader charge calculation for **g** ZIS structure, **h** D-ZIS structure, and **i** D-O-ZIS structure. The negative value is referred to obtain electrons, while the positive value means losing electrons. The structures are shown in top view. The bottom is a partial display of the corresponding structure. Source data are provided as a Source Data file.

Meanwhile, the O 1$s$ XPS spectrum after photocatalytic tests showed a signal of $^{\cdot}OOH$ in D-O-ZIS (Supplementary Fig. 34), further confirming the activity of overall water-splitting reactions in D-O-ZIS.

DFT calculations were performed to gain insight into the effects of distortion states and cation-site O doping. The calculated Bader charge over the structures revealed a strong charge redistribution in the regions of distortion states and O doping. In the ZIS structure, a charge transfer occurred from the Zn atom (0.861|e|) to the S atom (−0.805|e|) (Fig. 5g). In the D-ZIS structure, the charges on the $S_1$ and $S_2$ atoms were significantly redistributed (−0.194|e| for $S_1$ and −0.332|e| for $S_2$) in the dipolar bond of the $S_1$−$S_2$ center. Electrons were transferred from nearby Zn atoms (0.853|e|, 0.847|e|) to the $S_1$ and $S_2$ sites in D-ZIS, and the Zn vacancy acted as an electron trap (Fig. 5h)[51]. Cation-site O doping in D-O-ZIS resulted in a large electronegativity, causing the charge on the coordinated $S_2$ site to become more positive (1.024|e|), even higher than the charge on the Zn atom (0.865|e|). The charge on the $S_1$ site was −0.410|e|, indicating a significant difference in electronegativity between the adjacent sites (Fig. 5i). These findings are consistent with the XANES results of the O atom coordinated as S-O configuration (Fig. 2e). The additional charge transfer (from $S_2$ atoms to O atoms and electrons extracted at the Zn vacancy) improved charge mobility and generated a more positive charge center of the $S_2$ site, which may switch the active sites in the local structure.

To understand the role of active centers in catalytic processes, we investigated the distribution of charge density in the valence band maximum (VBM) and conduction band minimum (CBM)[23,28]. ZIS exhibited uniformly distributed charge densities in both the VBM and CBM (Fig. 6a). However, the VBM and CBM charge densities of D-ZIS were localized at the $S_1$−$S_2$ sites, Zn vacancy, and unsaturated S atoms (Fig. 6b). Meanwhile, the CBM charge density of D-O-ZIS was localized at the $S_1$−$S_2$−O sites and Zn vacancy, while the VBM charge density was significantly decreased at the $S_2$ sites. This indicates that electrons could be easily photoexcited to the conduction band, resulting in electron depletion at the $S_2$ site (Fig. 6c)[22]. Therefore, the photogenerated holes accumulation at the $S_2$ sites and electrons trap at the $S_1$ sites and Zn vacancy induced charges that were spatially separated within atomic sites, providing a strong driving force for efficient electron-hole separation on D-O-ZIS.

The partial density of states (PDOS) calculation was conducted to investigate the structure of distortion states and O doping in D-O-ZIS. The PDOS of D-O-ZIS showed S 3$p$-O 2$p$ bonding resonances near the Fermi level ($E_F$), with hybridized electronic states dominating the feature (Supplementary Fig. 35)[23]. The electronic states of the S 3$p$ and O 2$p$ increased at $E_F$, and the surrounding S atom was also activated compared to D-ZIS and ZIS in the $S_1$−$S_2$−O configuration of D-O-ZIS (Fig. 6d). D-O-ZIS showed a high degree of overlap for the intermediates (OH*, O*, and *OOH) adsorbed on the $S_2$ p-band, indicating strong interactions during the oxygen production process (Supplementary Fig. 36)[18]. The optimum hydrogen adsorption-free energy ($\Delta G_{H^*}$) was −0.07 eV at the $S_1$ site and -0.13 eV at the Zn vacancy, indicating that the $S_1$ site is more conducive to hydrogen adsorption/desorption than the Zn vacancy in D-O-ZIS, while D-ZIS showed an optimum $\Delta G_{H^*}$ of −0.63 eV at the $S_1$ site and ZIS of −1.10 eV at S site (Fig. 6e), and their adsorption models are depicted in Supplementary Fig. 37–39. The free energy barrier of O* adsorption at the $S_2$ site in D-O-ZIS was low (0.31 eV) compared to the energy barriers in ZIS (1.27 eV) and D-ZIS (0.97 eV), which enhances the oxygen evolution reaction (Supplementary Fig. 40). Furthermore, the $O_2$ evolution activity on metal atoms (i.e., Zn sites) was investigated in D-O-ZIS, excluding the role of metal sites as oxygen production sites (energy barrier 0.85 eV) in this photocatalyst design (Supplementary Fig. 41). By investigating structures with varying Zn vacancy levels and O doping (Supplementary Fig. 42), the D-O-ZIS significantly optimized the adsorption of key intermediates for $H_2$ and $O_2$ evolution (Supplementary Fig. 43). Consequently, D-O-ZIS exhibited hydrogen and oxygen species adsorbed at the $S_1$ and $S_2$ site, respectively, which would promote an overall water-splitting reaction.

We further performed the redox potential and free energy of sulfur ions oxidation by the photogenerated holes to elucidate the stability mechanism. The sulfur ions of ZIS or D-ZIS were easily oxidized and devitalized by photoinduced holes due to the redox potential of $S^{2-}/S^0$ (0.48 eV), while the sulfur ions in $S_1$−$S_2$−O had a lower redox potential of $S^1/S^0$ (0.42 eV) for D-O-ZIS with weak oxidation driving force (Supplementary Fig. 44). The energy barrier for sulfur ions oxidation in $S_1$−$S_2$−O configuration was 0.68 eV for D-O-ZIS, higher than that in D-ZIS (0.42 eV), and ZIS (0.37 eV), indicating that the sulfur ions in $S_1$−$S_2$−O configuration were difficult to oxidize by photogenerated holes (Fig. 6f). Additionally, the oxidation energy barrier of sulfur ions for D-O-ZIS (0.68 eV) was higher than the oxygen production free energy of 0.31 eV (Supplementary Fig. 40), which suggests that D-O-ZIS preferred to generate oxygen during water-splitting reactions instead of being oxidized by photogenerated holes.

We proposed the work principle of overall water-splitting on D-O-ZIS photocatalyst (Fig. 7), and the catalytic mechanisms of ZIS and D-ZIS were illustrated in Supplementary Fig. 45. Firstly, D-O-ZIS absorbs incident photon to produce photogenerated charge carriers. The photogenerated electron-hole pairs are efficiently separated driven by the internal electric field due to the strong dipole of the distortion-evoked cation-site O doping structure, and then transferred to the active $S_1$-$S_2$-O sites to undergo a redox reaction. Due to the optimized energy bands, the CB of D-O-ZIS is negative enough (−0.51 eV) to produce $H_2$, while the VB is positive enough (1.64 eV) to produce $O_2$. The strong charge redistribution character and large electronegativity differences between $S_1$ and $S_2$ atomic sites activate stable oxygen reactions at $S_2$ sites and avoids the common issue of sulfur instability in metal sulfide photocatalysis, while $S_1$ sites favor the adsorption/desorption of hydrogen. The co-catalysts of Pt and $CoO_x$ loading further enhance the photocatalytic activity of D-O-ZIS by promoting charges separation and reducing the free energy of hydrogen and oxygen adsorption. Consequently, D-O-ZIS as a single photocatalyst realizes efficient overall water splitting with high stability.

## Discussion

This work proposes an electronegativity difference strategy to activate and stabilize ZIS for photocatalytic overall water splitting, achieving a remarkable 0.57% solar-to-hydrogen conversion efficiency along with high stability. A distortion-evoked cation-site O doping in Zn atom sites of D-O-ZIS generates significant electronegativity differences between adjacent atomic sites, with $S_1$ sites being electron-rich and $S_2$ sites being electron-deficient in the local $S_1$−$S_2$−O structure. The strong charge redistribution character activates stable oxygen reactions at $S_2$ sites and hydrogen adsorption/desorption at $S_1$ sites. Our study showcases the universal applicability of activating and stabilizing metal sulfides photocatalysts, such as ZIS, $MoS_2$ and $In_2S_3$, for efficient photocatalytic overall water splitting through distortion-evoked cation-site O doping strategy from the perspective of electronegativity differences.

## Methods

### Synthesis of photocatalysts

Synthesis of $ZnIn_2S_4$ (ZIS). The synthesis of ZIS typically involves the following procedure:[7] 1 mmol of $ZnCl_2$, 2 mmol of $In(NO_3)_3$, and 4 mmol of thioacetamide were dissolved in 35 mL of deionized water and stirred vigorously for 30 min. The mixed solution was then transferred to a 50 mL Teflon-lined autoclave and heated at 180 °C for 12 h. After cooling to room temperature, the resulting yellow suspension was collected and washed with ethanol and deionized water four times, respectively. Finally, the product was dried at 60 °C overnight before further use.

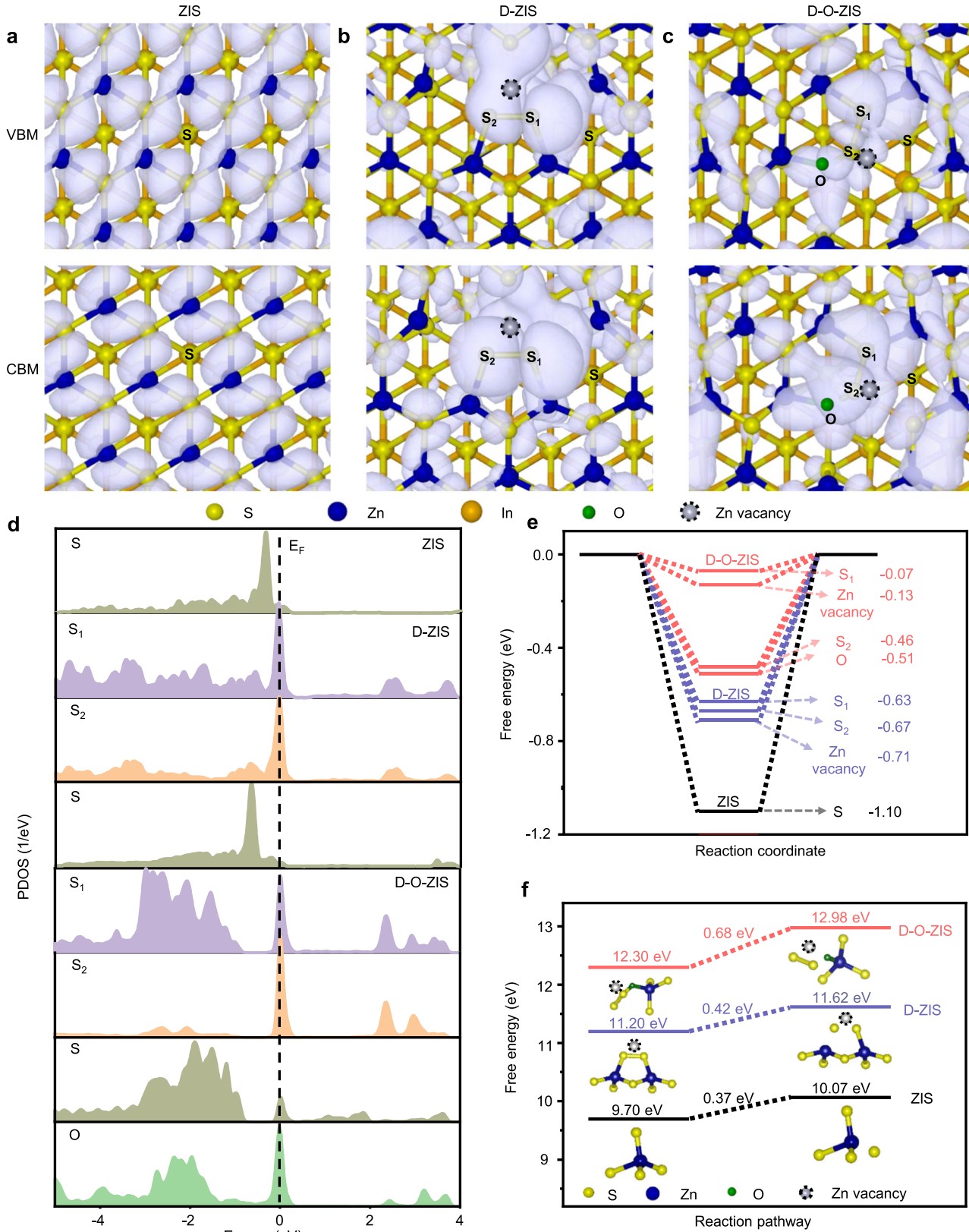

**Fig. 6 | Photocatalytic overall water-splitting working principle determined by DFT.** Distribution of partial charge density near the edge of conduction band and valence band of **a** ZIS, **b** D-ZIS, and **c** D-O-ZIS. The iso-surface value is 0.015 e Å⁻³. The structures are shown in top view. **d** PDOS of different S atoms of $S_1$, $S_2$, surrounding normal S, and O atoms in D-O-ZIS, and $S_1$ and $S_2$ atoms in D-ZIS, and S atom in ZIS; Fermi level ($E_F$); **e** The computed values of $\triangle G_{H^*}$ at different sites in ZIS, D-ZIS, and D-O-ZIS; **f** The free energy of sulfur ions oxidation in ZIS, D-ZIS, and D-O-ZIS. The included configurations are the S atom leaching by photogenerated holes oxidation from pristine structure. Source data are provided as a Source Data file.

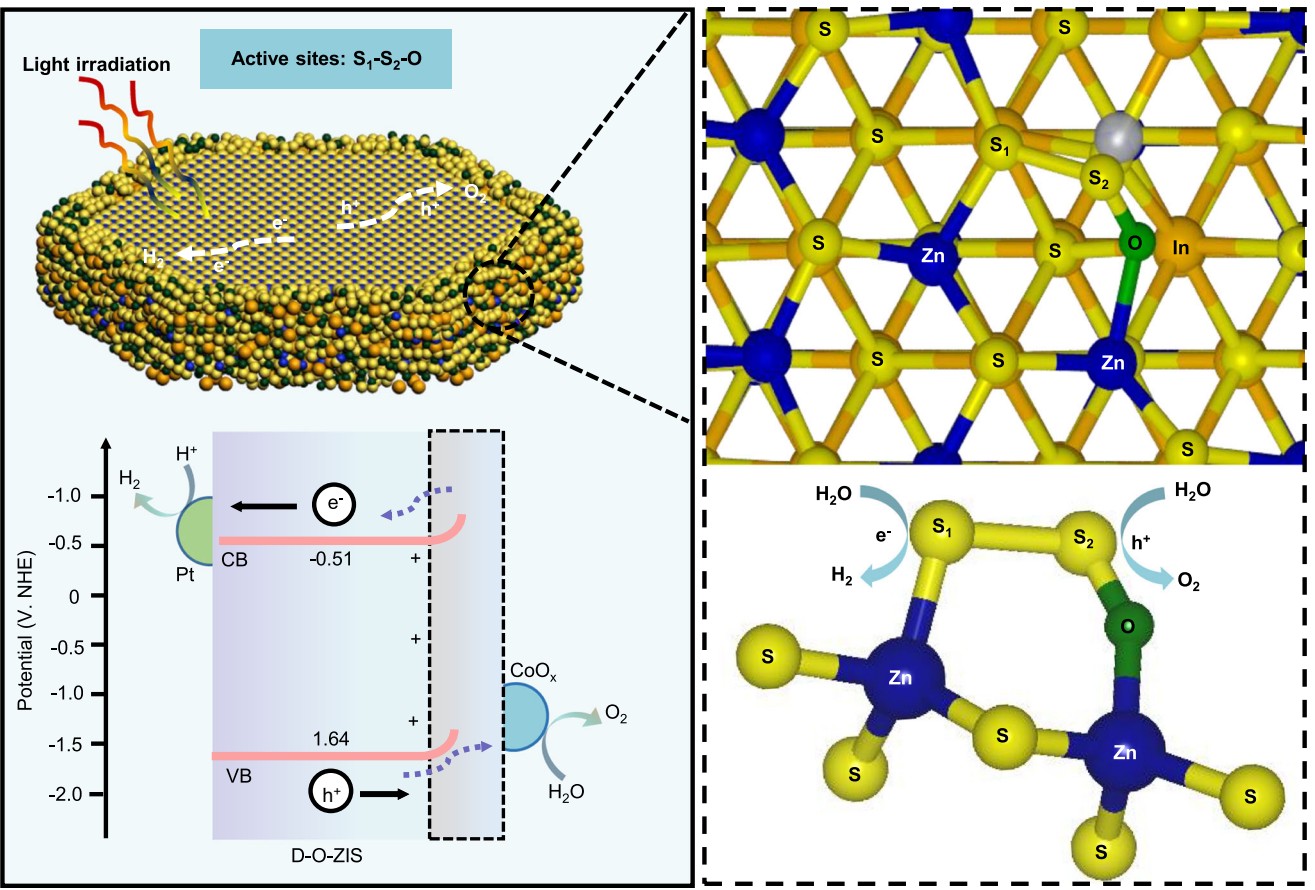

**Fig. 7 | Photocatalytic overall water-splitting working principle.** The local structure is shown in top view, and the bottom is a partial display of the corresponding structure. Conduction band (CB), Valence band (VB). Source data are provided as a Source Data file.

**Synthesis of distorted ZIS (D-ZIS).** The D-ZIS was synthesized starting from ZIS, using a thermal migration strategy. The synthesis process involved heating the precursor material of ZIS at 500 °C for 30 min under an atmosphere of Ar/H₂ (1 bar). Once the reaction was completed, the resulting powder product was allowed to cool naturally to room temperature, after which it was collected and washed several times using ethanol and deionized water. The product was then dried at 60 °C under vacuum overnight to ensure complete removal of any remaining solvent before further analysis.

Synthesis of distortion-evoked cation-site oxygen doping of ZIS (D-O-ZIS). For the synthesis of D-O-ZIS, the resulting D-ZIS was treated with Ar/O₂ (5%) flow at 500 °C for 10 min. The resulting powder was then collected and washed three times with ethanol and deionized water, respectively, before being dried under vacuum at 60 °C.

**Photocatalytic overall water-splitting reaction test**

To perform photocatalytic reactions, we used a reaction vessel with a gas-closed circulation and evacuation system. Before each reaction, we dispersed 35 mg of photocatalysts in 50 mL of pure water and evacuated the air from the vessel, replacing it with Ar gas. We then conducted photocatalytic H₂ and O₂ evolution in a quartz reactor using a 300 W Xe lamp for irradiation. The evolved gases were pumped and detected by a Shimadzu GC-2014c gas chromatography with a thermal conductivity detector. We measured the STH efficiency under simulated sunlight at AM1.5 G illumination (100 mW cm⁻²). The STH efficiency was determined according to the following Eq. (1):

$$STH(\%) = (R(H_2) \times \Delta G_r)/(P \times S) \times 100\% \tag{1}$$

Here, $R(H_2)$, $\Delta G_r$, $P$, and $S$ denote the H₂ evolution rate, the reaction Gibbs energy during the water-splitting reaction, the light energy flux under the AM1.5 G irradiation, and the irradiated sample area, respectively. The value $\Delta G_r$ used for the calculations is 237 kJ mol⁻¹ for the liquid water in the reaction system. The value of $P$ is 100 mW cm⁻². The value S is 3.6 cm².

## Data availability

All data generated in this study are provided in the article and Supplementary Information, and the raw data generated in this study are provided in the Source Data file. Source data are provided with this paper.

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

## Acknowledgements

This research is supported by the National Natural Science Foundation of China (22261142666, 52172237), the Shaanxi Science Fund for Distinguished Young Scholars (2022JC-21), the Research Fund of the State Key Laboratory of Solidification Processing (NPU), China (Grant No. 2021-QZ-02), and the Fundamental Research Funds for the Central Universities (3102019JC005, D5000220033). All fundings are awarded to X.L.

## Author contributions

X.X. and X.L. conceived the idea. X.L. supervised the project. X.X. performed the synthesis, characterization, and photocatalysis. Y.L., Y.Z., and Y. Wang performed the photo-electrochemical experiments. X.C., Y. Wei, C.D., J.S., R.W., P.G., J.Y., J.Z., A.J.S., and M.-M.T. analyzed the data and commented on the manuscript.

## Competing interests

The authors declare no competing interests.

## Additional information

[1]State Key Laboratory of Solidification Processing, Center for Nano Energy Materials, School of Materials Science and Engineering, Northwestern Polytechnical University, Xi'an 710072, China. [2]Research & Development Institute of Northwestern Polytechnical University, Shenzhen 518057, China. [3]Institute of High Performance Computing (IHPC), Agency for Science, Technology and Research (A*STAR), 1 Fusionopolis Way, #16-16 Connexis, Singapore 138632, Singapore. [4]Department of Physics, National University of Singapore, Singapore 117576, Singapore. [5]College of Science, Gansu Agricultural University, Lanzhou 730070, China. [6]Singapore Synchrotron Light Source, National University of Singapore, 5 Research Link, Singapore 117603, Singapore. [7]College of Microelectronics, Xidian University, Xi'an 710072, China. [8]School of Engineering and Materials Science, Faculty of Science and Engineering, Queen Mary University of London, Mile End Road, London E1 4NS, UK. [9]Department of Chemical Engineering, Imperial College London, South Kensington Campus, London SW7 2AZ, UK. [10]These authors contributed equally: Xu Xin, Yuke Li. ✉e-mail: lixh32@nwpu.edu.cn

