## [Peer Review File · Nature Communications]

REVIEWER COMMENTS

Reviewer #1 (Remarks to the Author):

Xu et al. synthesized an excellent Zn_{1-x}S₄-based catalyst for the synthesis of H₂ and O₂ from photocatalytic water. They optimized the water splitting activity of Zn_{1-x}S₄ by doping O at the cation site position, which provides a new perspective on the activation and stabilization of metal sulfide photocatalysts for efficient overall water-splitting. This is a critical issue, and the work, in my opinion, deserves to be published. This work might be significantly improved if the author considers the following points:

1. From the photocatalytic activity results (Figure 3d) of ZIS, D-ZIS, D-O-ZIS, all have catalytic activity to produce hydrogen, but only oxygen production activity occurs on the surface of D-O-ZIS. And their structural differences are the concentration of Zn vacancy and the concentration of doped O atom. It appears that increasing the Zn vacancy concentration can increase the hydrogen evolution activity. Have the effects of different Zn vacancy and doped O concentration on the catalytic activity been considered? The reviewer suggests that the authors should consider the effect of concentration (Zn vacancies and doped O atoms) on catalytic activity and stability.
2. The authors believe that the surface is doped with O, but no Zn-O bond signal is found from the results of XPS and Raman spectroscopy. Is it possible that the O atom is surface bound to the S atom and not doped between the Zn-S atoms as in Figure 2g?
3. A full description of the photocatalytic process does not appear in the article, which makes it difficult for the reader to understand the generation of photoactivated electrons (holes) at the catalytic interface, which then catalyze the photocatalytic splitting of water molecules. The authors should provide such an explanation or illustrate the reaction process with a diagram.
4. In the section 'Kinetics of charge transport and separation', the authors demonstrate that the internal electric field drives the charge distribution at the catalytic interface. However, there is no close relationship between them, let alone that the internal electric field is the result of charge redistribution. It seems to me that the effect of the internal electric field is more likely to affect the adsorption and desorption of small molecules such as H₂O, H, OH, O and OOH. The authors should revise this section.
5. On the surface of ZIS or D-ZIS, the photolytic mechanism is the decomposition of water into H₂, so what is the final destination of the O in H₂O and does the O oxidize the S-S bond and form an active center similar to that of D-O-ZIS? The catalytic mechanism of ZIS or D-ZIS should be stated in the paper.
6. The author stated the adsorption strength of the Zn site and the S site for H (line 268), but in terms of adsorption energy, it appears that the Zn site is the most stable adsorption site for H, rather than the S site (principle of minimum energy). The author should rearrange the relevant calculation results and expressions.

Reviewer #2 (Remarks to the Author):

Although this manuscript describes a new type of OWS photocatalyst and shows H₂ and O₂ evolution data, the reviewer finds several unreasonable or strange data and interpretations as follows:

1. Overall, I find the performance data shown in this work very strange. First of all, Pt is very active for the reverse reaction (HOR and ORR). An amorphous overlayer such as chromium oxyhydroxide to cap the noble metal is critical for inorganic semiconductor-based photocatalysts to split water, as demonstrated by many studies of this field. In this regard, I have doubts that the Pt/CoOx cocatalyst used in this work will absolutely cause reverse reactions. The authors checked ORR using the bare sample in Figure S12, which is meaningless. However, the authors did not perform reverse reaction examination over the Pt/CoOx-modified photocatalyst, which should be critically checked. Secondly, in the performance figures such as Figure S8, S14, S18a, there is a significant deactivation between adjacent runs; however, there is no deactivation within each run (the line is so straight within each run). This is impossible! And in these cases, almost only hydrogen was produced, what were the oxidation products resulting from holes?

2. Cocatalysts decoration methods should have been given in detail, as they are very important components for a photocatalyst. The structure, dispersion and chemical states of each cocatalyst component should be analyzed. However, these data are unexpectedly missing. And why was the activity almost the same by loading significantly different ratios of Pt/CoOx (1:4 and 3:1) in Figure S7a and b?

3. The weight-dependent test showed a saturated rather than linear increased activity with the increased amount of photocatalyst (Figure S9b). Therefore, it is unreasonable to normalize the activity with gram in Figure 3d and Figure S7c.

4. According to the equation of $\text{IO}_3^- + 6\text{e}^- + 3\text{H}_2\text{O} = \text{I}^- + 6\text{OH}^-$, 50 mL of 20 mM NaIO₃ aqueous contains 1 mmol NaIO₃, which consumes 6 mmol of electron and therefore 1.5 mmol of O₂ can be evolved. Therefore, it's impossible to produce ~1.6 mmol O₂ in the first 12 hours and continue to consume electrons after 12 hours in Figure S15c.

5. Some characterization data are not correctly interpreted. For example, the EPR signal at g value of 2.004 was assigned by the authors to zinc vacancies. However, this signal, to be correct, should be assigned to free electrons or unpaired electrons trapped in a vacancy. Individual zinc vacancies will not produce an EPR signal, and in my opinion, the detected EPR signal at g value of 2.004 probably results from unpaired electrons trapped in anion (O or S) vacancies. ICP is suggested to study the loss of zinc.

As a whole, I cannot recommend publication of this work in Nature Communications.

Reviewer #3 (Remarks to the Author):

This manuscript proposes an electronegativity difference strategy to activate and stabilize ZIS for photocatalytic overall water-splitting, achieving a remarkable 0.57% solar-to-hydrogen conversion efficiency along with high stability. A distortion-evoked cation-site O doping in Zn atom sites of D-O-ZIS generates significant electronegativity differences between adjacent atomic sites, with S1 sites being electron-rich and S2 sites being electron-deficient in the local S1–S2–O structure. The strong charge redistribution character activates stable oxygen reactions at S2 sites and hydrogen

adsorption/desorption at S1 sites. This is a carefully done study and the findings are of considerable interest. A few minor revision are list below.

1. How to determine the content of zinc vacancies by Zn XPS spectra (Figure S3c)? The author need to provide the depth explanation.
2. Why does the incorporation of O atoms lead to the formation of more Zn vacancies ? To further prove it, it is recommended that the author provide comparative experimental results.
3. There is a lack of comparison for AQY and STH of D-O-ZIS and other ZnIn₂S₄ based nanocomposites for overall water splitting. The author should add it and discuss in more detail.
4. The photocatalytic H₂ and O₂ evolution amounts up to 373.2 and 178.8 μmol , the ratio of H₂ and O₂ exceeds 2:1, why is O₂ evolution amounts so high ?
5. In order to confirm the stability of the sample, the author is requested to provide the XRD, XPS, SEM, TEM and HRTEM images of the sample after the photocatalytic reaction.

Point-by-point Response to the Reviewers' Comments

Reviewer #1 (Remarks to the Author):

*Xu et al. synthesized an excellent ZnIn₂S₄-based catalyst for the synthesis of H₂ and O₂ from photocatalytic water. They optimized the water splitting activity of ZnIn₂S₄ by doping O at the cation site position, which provides a new perspective on the activation and stabilization of metal sulfide photocatalysts for efficient overall water-splitting. **This is a critical issue, and the work, in my opinion, deserves to be published.** This work might be significantly improved if the author considers the following points:*

Response: We thank the referee for appreciating the impact and value of our study. We also appreciate the referee's constructive comments, which have helped us improve the quality of our manuscript. We have revised the manuscript accordingly.

1. From the photocatalytic activity results (Figure 3d) of ZIS, D-ZIS, D-O-ZIS, all have catalytic activity to produce hydrogen, but only oxygen production activity occurs on the surface of D-O-ZIS. And their structural differences are the concentration of Zn vacancy and the concentration of doped O atom. It appears that increasing the Zn vacancy concentration can increase the hydrogen evolution activity. Have the effects of different Zn vacancy and doped O concentration on the catalytic activity been considered? The reviewer suggests that the authors should consider the effect of concentration (Zn vacancies and doped O atoms) on catalytic activity and stability.

Response: Thanks very much for the valuable comments. We explored theoretical structural models with varying concentrations of Zn vacancies or O dopants to assess their formation energies, as well as the hydrogen/oxygen adsorption energies on these sites, determining the activity and stability of the structures. A single Zn vacancy formation proves energetically favorable in constructing distorted structures. O doping benefits the structure, although introducing more than three oxygen atoms necessitates substantial energy input. Additionally, O doping significantly enhances hydrogen adsorption while minimally affecting oxygen adsorption.

We built ZIS models with different Zn vacancy concentrations ranging from one to four vacancies within a crystal unit as shown in **Figure R1a**. Starting from ZIS, the formation of a single vacancy accompanied by the formation of an S-S bond requires an energy input of 5.99 eV. When two vacancies

are formed, a higher energy input of 6.73 eV is required. As the Zn vacancy concentration continues to increase, the energy input needed is 2.24 eV. Therefore, according to the principle of minimum energy, the ZIS structure tends to favor the formation of a stable structure of D-ZIS structure, which consists of a single vacancy and an accompanying S-S bond. However, as the Zn vacancy concentration further increases, the barrier for structure formation becomes higher, making it increasingly difficult to form.

Figure R1. DFT calculations on structures with varying Zn vacancy levels and O doping. **a** Structural transition models and formation energies at different Zn vacancy concentrations; **b** Structural transition models and formation energies at different O doped concentrations; **c** The computed values of ΔG_{H^*} at different sites in ZIS, Zn vacancy-1, 2, and 3, and O doping-1, 2, and 3; **d** DFT calculated free energy profile of OER process on O doping-1, 2, and 3 at pH=0 and U=1.23 V vs. SHE.

We investigated various concentrations of O doping based on D-ZIS structure in **Figure R1b**. When a single O dopant is introduced, the energy required for structure formation is -3.45 eV, indicating that O doping is energetically favorable. As the concentration of O doping increases (with two oxygen dopants), the formation energy decreases to -0.43 eV. However, with further increases in the O doping concentration, the energy barrier for structure formation becomes 0.23 eV, making it

relatively challenging.

Furthermore, **Figure R1c, d** illustrates the hydrogen adsorption energy (ΔG_{H^*}) for different structures, including various Zn vacancy structures and O-doped structures, as well as the oxygen adsorption barriers for the O-doped structures to investigate their catalytic activity. It can be observed that the optimal hydrogen adsorption occurs at the S site when a single Zn vacancy concentration or two Zn vacancy concentrations are present, with corresponding values of -0.63 eV and -0.72 eV, respectively. As the Zn vacancy concentration continues to increase, the adsorption weakens, reaching -0.88 eV. For a single O doping concentration, the optimal hydrogen adsorption energy at the S site is -0.08 eV, while with two or three O dopants, the hydrogen adsorption energy is -0.05 and -0.18 eV, respectively (**Figure R1c**). These results indicate that O doping enhances the electronic state of the S site, leading to favorable hydrogen adsorption. **Figure R1d** demonstrates that varying O doping concentrations have a minimal effect on the catalyst's oxygen evolution barriers. The oxygen evolution barriers for one, two, and three O doping concentrations are calculated to be 0.31 eV, 0.33 eV, and 0.35 eV, respectively.

In the revised version, **Figure R1** is added as new **Figure S40** in the supporting information. The relative discussion has been revised in the page 13 of main text, page 22 of supporting information, and copied below:

“..... (Figure S39). DFT calculations further confirmed the exceptional activity and superiority of the D-O-ZIS configuration by studying structures with varying Zn vacancy levels and O doping (Figure S40).” (Page 14 of main text)

“As shown in Figure S40a, the formation of a single vacancy accompanied by the formation of an S-S bond requires an energy input of 5.99 eV. When two vacancies are formed, a higher energy input of 6.73 eV is required. As the Zn vacancy concentration continues to increase, the energy input needed is 2.24 eV. Therefore, according to the principle of minimum energy, the ZIS structure tends to favor the formation of a stable structure of D-ZIS structure, which consists of a single vacancy and an accompanying S-S bond. However, as the Zn vacancy concentration further increases, the barrier for structure formation becomes higher, making it increasingly difficult to form. We conducted an investigation into various concentrations of O doping using calculations based on D-ZIS structure in Figure S40b. When a single oxygen dopant is introduced, the energy required for structure formation is -3.45 eV, indicating that O doping is energetically favorable. As the concentration of O doping increases (with two oxygen dopants), the formation energy decreases to -0.43 eV. However, with further increases in the O doping concentration, the energy

barrier for structure formation becomes 0.23 eV, making it relatively challenging.

As shown in **Figure S40c, d**, it can be observed that the optimal hydrogen adsorption occurs at the S site when a single Zn vacancy concentration or two Zn vacancy concentrations are present, with corresponding values of -0.63 eV and -0.72 eV, respectively. As the Zn vacancy concentration continues to increase, the adsorption weakens, reaching -0.88 eV. For a single O doping concentration, the optimal hydrogen adsorption energy at the S site is -0.08 eV, while with two or three oxygen dopants, the hydrogen adsorption energy is -0.05 and -0.18 eV, respectively (**Figure S40c**). These results indicate that O doping enhances the electronic state of the S site, leading to favorable hydrogen adsorption. **Figure S40d** demonstrates that varying O doping concentrations have a minimal effect on the catalyst's oxygen evolution barriers. The oxygen evolution barriers for one, two, and three O doping concentrations are calculated to be 0.31 eV, 0.33 eV, and 0.35 eV, respectively." (Page 67 of supporting information)

2. The authors believe that the surface is doped with O, but no Zn-O bond signal is found from the results of XPS and Raman spectroscopy. Is it possible that the O atom is surface bound to the S atom and not doped between the Zn-S atoms as in Figure 2g?

Response: Thanks for your comments. Oxygen atoms are mainly doped at the cationic positions of D-O-ZIS, forming S-O and a small amount of Zn-O bond structures. We have re-labeled the Zn K-edge EXAFS spectrum, as shown in the **Figure R2a**. The Fourier transformed curve of O-doped D-O-ZIS shows not only the Zn-S coordination with a main peak at 1.90 Å, but also a weak peak at about 1.39 Å assigned to the Zn-O coordination (*Angew. Chem. Int. Ed.* 2016, 55, 6716). We have re-peak-fitted the O 1s XPS spectrum of oxygen lattice, as shown in **Figure R2b**. The peak at 529.81 eV is ascribed to the Zn-O bond (*Journal of Environmental Chemical Engineering*, 2022, 10, 108587). The higher binding energy shifts of Zn 2p peaks in D-ZIS (0.2 eV shift) and D-O-ZIS (0.4 eV shift) suggested increased electron loss of Zn atoms due to edge distortion or O atom doping incorporation (i.e. **Figure S3c**). In addition, the S-O coordinated bond is confirmed by the S K-edge XANES spectra at 2481.8 eV and the corresponding higher shifts of rising S pre-edge (i.e. **Figure 2e**). Thus, the doped oxygen atoms coordinate with both the Zn atoms and the S atoms. DFT calculation simulations of distortion states and O doping structures (i.e. **Figure 2g**) revealed that the lattice parameters matched the experimental structural results.

Figure R2. Geometric and local electronic structures of photocatalysts. **a** The Fourier transform curves of k^3 -weighted Zn K-edge EXAFS spectra of ZIS, D-ZIS, and D-O-ZIS; **b** XPS spectra of O 1s in ZIS, D-ZIS and D-O-ZIS.

In the revised version, **Figure R2a** is added as new **Figure 2c** in the main text. **Figure R2b** is added as new **Figure S3a** in the supporting information. The relative discussion has been added in the page 7 of main text, page 20 of supporting information, and copied below:

*“.....with a concentration of ~3.7% (**Figure S3c**). A weak peak at about 1.39 Å assigned to the Zn-O coordination in D-O-ZIS [24].” (Page 7 of main text)*

“In the lattice oxygen peak, Zn-O bonding can be observed at 529.81 eV [19].” (Page 20 of supporting information)

3. A full description of the photocatalytic process does not appear in the article, which makes it difficult for the reader to understand the generation of photoactivated electrons (holes) at the catalytic interface, which then catalyze the photocatalytic splitting of water molecules. The authors should provide such an explanation or illustrated the reaction process with a diagram.

Response: Thanks very much for the valuable comments. Actually, we have discussed the work principle of overall water-splitting on D-O-ZIS photocatalyst in Figure 7 of the original version. In the revised version, we have re-strengthened the elaboration of the photocatalytic process in the revised text.

*“We proposed the work principle of overall water-splitting on D-O-ZIS photocatalyst (**Figure 7**)....Firstly,..... and then transferred to the active S_1 - S_2 -O sites to undergo a redox reaction. Due to the optimized energy bands, the CB of D-O-ZIS is negative enough (-0.51 eV) to produce H_2 , while the VB is positive enough (1.64 eV) to produce O_2 . The strong charge redistributionwhile S_1 sites favor the adsorption/desorption of hydrogen.....” (Page 14 of main text)*

4. In the section “Kinetics of charge transport and separation”, the authors demonstrate that the internal electric field drives the charge distribution at the catalytic interface. However, there is no close relationship between them, let alone that the internal electric field is the result of charge redistribution. It seems to me that the effect of the internal electric field is more likely to affect the adsorption and desorption of small molecules such as H_2O , H , OH , O and OOH . The authors should revise this section.

Response: Thanks very much for the valuable comments. We have corrected the description regarding the internal electric field drives the charge distribution at the catalytic interface in the section of “Kinetics of charge transport and separation” in the revised text. Furthermore, we employed DFT calculations to confirm the impact of internal electric field on the adsorption behavior of crucial H and OH intermediates.

To shed more light on the effects of diverse intensities of built-in electric field, DFT calculation was performed to investigate the adsorption of reaction intermediates on photocatalysts and cocatalysts (**Figure R3**). The hydrogen adsorption free energy (ΔG_{H^*}) on ZIS, D-ZIS, and D-O-ZIS was firstly obtained, as shown in **Figure R3a**. The ZIS reveals the lowest adsorption energy of -1.10 eV. The low value demonstrates strong hydrogen adsorption, which indicates a relatively difficult desorption process. The ΔG_{H^*} of D-ZIS is slight negative with a value of -0.63 eV, which means an improved adsorption process. Compared with D-ZIS, the ΔG_{H^*} of D-O-ZIS is significantly improved from -0.63 to -0.07, demonstrating a moderate adsorption strength.

To further investigate the influence of the internal electric field on hydrogen adsorption, we examined the hydrogen adsorption on different Pt sites of ZIS/Pt, D-ZIS/Pt, and D-O-ZIS/Pt, as depicted in **Figure R3b**. The catalyst and Pt form a Schottky structure, and there is the largest Fermi level difference between D-O-ZIS and Pt, with large built-in electric field strength, and the calculated D-O-ZIS/Pt has the optimal hydrogen adsorption of 0.06 eV. This implies the charge injection and charge depletion of photocatalysts influenced by the introduction of strong internal electric field (**Figure R3c**).

Accordingly, the theoretical simulations verify that the enhanced internal electric field plays a crucial role in engineering OH^* adsorption strength, thus regulating the hydroxide adsorption processes (**Figure R3d**). The relationship indicates an optimal value of ΔG_{OH^*} of D-O-ZIS, which should be slightly negative. The biased adsorption energy may hinder either the hydroxide desorption

process, causing the site blocking effect. The electron transfer in D-O-ZIS, D-O-ZIS/Pt, and D-O-ZIS/CoO_x driven by the internal electric field modify the electronic states of photocatalyst surface, significantly optimizing the adsorption of key intermediates and endowing D-O-ZIS with highly active oxygen evolution reaction sites (*Angew. Chem. Int. Ed.* 2022, 61, e202208642, *Angew. Chem. Int. Ed.* 2022, 61, e202116057). Thus, the hydrogen/oxygen adsorption confirms the presence of charges injection and depletion of the D-O-ZIS induced by the internal electric field, thereby stimulating hydrogen and oxygen activities of D-O-ZIS.

Figure R3. The impact of the internal electric field on the adsorption behavior. **a** Gibbs free energies changes of HER process on S sites of ZIS, D-ZIS, and D-O-ZIS; **b** Gibbs free energies changes of HER process on Pt sites of ZIS/Pt, D-ZIS/Pt, and D-O-ZIS/Pt; **c** Electron injection/depletion effect of S sites and Pt sites on the ΔG_{H^*} ; **d** Adsorption free energies of OH* on ZIS, D-ZIS, and D-O-ZIS and on ZIS/CoO_x, D-ZIS/CoO_x, and D-O-ZIS/CoO_x at pH=0 and U=1.23 V vs. SHE.

In the revised version, the description regarding the internal electric field drives the charge distribution has been revised in the page 11 of main text. **Figure R3** is added as new **Figure S41** in the supporting information; and the relevant discussion has been added in the pages 14 of main text and 69 of supporting information, and copied below:

“.....Further analysis of the dipole moment, inducing a dipole of structure and adding asymmetry to the local structure, thus increasing the internal electric field (**Figure 4e**).” (Page 11 of main text)

“DFT calculations further....(Figure S40)....In addition, the internal electric field also plays a crucial role in the adsorption process (Figure S41).” (Page 14 of main text)

“Furthermore, we employed DFT calculations to investigate the impact of the internal electric field on the adsorption behavior of crucial H and OH intermediates (Figure S41). The hydrogen adsorption free energy (ΔG_{H^}) on ZIS, D-ZIS, and D-O-ZIS was firstly obtained, as shown in Figure S41a. The ZIS reveals the lowest adsorption energy of -1.10 eV. The low value demonstrates strong hydrogen adsorption, which indicates a relatively difficult desorption process. The ΔG_{H^*} of D-ZIS is slight negative with a value of -0.63 eV, which means an improved adsorption process. Compared with D-ZIS, the ΔG_{H^*} of D-O-ZIS is significantly improved from -0.63 to -0.07, demonstrating a moderate adsorption strength. To further investigate the influence of the internal electric field on hydrogen adsorption, we examined the hydrogen adsorption on different Pt sites of ZIS/Pt, D-ZIS/Pt, and D-O-ZIS/Pt, as depicted in Figure S41b. The catalyst and Pt form a Schottky structure, and there is the largest Fermi level difference between D-O-ZIS and Pt, with large built-in electric field strength, and the calculated D-O-ZIS/Pt has the optimal hydrogen adsorption of 0.06 eV. This implies the charge injection and charge depletion of photocatalysts influenced by the introduction of strong internal electric field (Figure S41c).*

“Accordingly, the theoretical simulations verify that the enhanced internal electric field plays a crucial role in engineering OH^ adsorption strength, thus regulating the hydroxide adsorption processes (Figure S41d). The relationship indicates an optimal value of ΔG_{OH^*} of D-O-ZIS, which should be slightly negative. The biased adsorption energy may hinder either the hydroxide desorption process, causing the site blocking effect. The electron transfer in D-O-ZIS, D-O-ZIS/Pt, and D-O-ZIS/ CoO_x driven by the internal electric field modifies the electronic states of photocatalyst surface, significantly optimizing the adsorption of key intermediates and endowing D-O-ZIS with highly active oxygen evolution reaction sites. Thus, the hydrogen/oxygen adsorption confirms the presence of charges injection and depletion of the D-O-ZIS induced by the internal electric field, thereby stimulating hydrogen and oxygen activities of D-O-ZIS.” (Page 69 of supporting information)*

5. On the surface of ZIS or D-ZIS, the photolytic mechanism is the decomposition of water into H_2 , so what is the final destination of the O in H_2O and does the O oxidize the S-S bond and form an active center similar to that of D-O-ZIS? The catalytic mechanism of ZIS or D-ZIS should be stated in the paper.

Response: Thank you very much for the valuable comments. In the case of ZIS or D-ZIS, the final destination of the oxygen in H_2O is the formation of SO_4^{2-} in the solution, which leads to the leaching

of S from ZIS and D-ZIS. The catalytic mechanism of ZIS or D-ZIS is added in the revised supporting information.

The oxidation of the S-S bond does not form an active center similar to D-O-ZIS. Instead, it damages the structure of the catalyst, resulting in the presence of oxidized sulfur ions in the solution. The photo-corrosion of photocatalysts for ZIS and D-ZIS induced by S leaching after the photocatalytic reaction was confirmed by S 2p XPS (i.e. **Figure 3e**). The S 2p spectra of ZIS and D-ZIS showed lower binding energy shifts of 0.62 and 0.59 eV, respectively, with reduced intensities, indicating that S leaching in ZIS and D-ZIS occurred and deactivating the catalyst. We have confirmed the presence of the small bump peak corresponding to SO_4^{2-} in the S 2p spectrum after the reaction as depicted in **Figure 4a**, and we have redrawn the graph accordingly in the main text. The binding energy peak of SO_4^{2-} at 169.89 eV is observed in both ZIS and D-ZIS after the reaction (*Adv. Energy Mater.* 2021, 11, 2101181). This finding suggests that the oxygen present during the water splitting process of ZIS and D-ZIS disrupts the atomic structure of S, leading to the formation of SO_4^{2-} ions adsorbed on the material surface. The HRTEM image of ZIS and D-ZIS after testing also revealed S leaching features, forming S-hole structure and the shell thickness is destroyed (**Figure R4b, c**).

Figure R4. Photocatalytic performance. **a** The S 2p XPS spectra of ZIS and D-ZIS before and after 120 h photocatalytic test. The vertical bars indicate the difference in intensity before and after test; HRTEM of ZIS and D-ZIS after photocatalytic testing: **b** ZIS, **c** D-ZIS.

The catalytic mechanisms of ZIS and D-ZIS were illustrated in **Figure R5**. Firstly, ZIS and D-ZIS absorb incident photon to produce photogenerated charge carriers. The photogenerated electron-

hole pairs are separated, and then transferred to the active sites to undergo a redox reaction. Due to the optimized energy bands, the CB of ZIS (-0.43 eV) and D-ZIS (-0.46 eV) is negative enough to produce H₂. Although the valence band (VB) energy is sufficiently positive to enable O₂ production in ZIS and D-ZIS, the inherent instability of sulfur atoms in these materials makes them prone to oxidation by holes. This leads to the generation of sulfate ions in the solution and, consequently, the degradation of the catalysts.

Figure R5. Photocatalytic water splitting mechanisms of ZIS and D-ZIS.

In the revised version, **Figure R4a** has been revised in **Figure 3e** in the main text. **Figure R4b, c** are added as new **Figure S22** in the supporting information. **Figure R5** is added as new **Figure S43** in the supporting information; and the relative discussion has been revised in the page 9, 14 of main text, page 51, 71 of supporting information, and copied below:

“The S 2p spectra of D-O-ZIS showed with reduced intensities and oxidation product of SO₄²⁻, indicating that S leaching.....The HRTEM image and structural analysis of D-O-ZIS....., while ZIS and D-ZIS exhibited S leaching characteristics (Figure S22)”. (Page 9 of main text)

“We proposed (Figure 7), and the catalytic mechanisms of ZIS and D-ZIS were illustrated in Figure S43.” (Page 14 of main text)

“ZIS and D-ZIS suffer from corrosion under light irradiation, during which lattice S²⁻ ions are more easily escaped and oxidized by photoexcited holes into SO₄²⁻, thus leading to deactivation of photocatalysts. The HRTEM image of ZIS and D-ZIS after testing revealed S leaching features, forming S atom loss structures and the shell thickness for D-ZIS is destroyed.” (Page 51 of supporting information)

“The catalytic mechanisms of ZIS and D-ZIS were illustrated in Figure S43. Firstly, ZIS and D-ZIS absorb incident photon to produce photogenerated charge carriers. The photogenerated electron-hole pairs are separated, and then transferred to the active sites to undergo a redox reaction. Due to the optimized energy bands, the CB of

ZIS (-0.43 eV) and D-ZIS (-0.46 eV) is negative enough to produce H_2 . Although the valence band (VB) energy is sufficiently positive to enable O_2 production in ZIS and D-ZIS, the inherent instability of sulfur atoms in these materials makes them prone to oxidation by holes. This leads to the generation of sulfate ions in the solution and, consequently, the degradation of the catalysts.” (Page 71 of supporting information)

6. The author stated the adsorption strength of the Zn site and the S site for H (line 268), but in terms of adsorption energy, it appears that the Zn site is the most stable adsorption site for H, rather than the S site (principle of minimum energy). The author should rearrange the relevant calculation results and expressions.

Response: Thanks very much for your comments. Maybe there is a misunderstanding. The optimum hydrogen adsorption-free energy (ΔG_{H^*}) was -0.07 eV at the S_1 site (**Note: not S_2 site**) and -0.13 eV at the Zn vacancy. The adsorption energy for S_1 site is more moderate and closer to zero (the ideal adsorption energy) compared to Zn site, indicating that the S_1 site is more conducive to hydrogen adsorption/desorption than the Zn vacancy in D-O-ZIS.

Reviewer #2 (Remarks to the Author):

Although this manuscript describes a new type of OWS photocatalyst and shows H₂ and O₂ evolution data, the reviewer finds several unreasonable or strange data and interpretations as follows:

Response: We appreciate the referee's constructive comments, which have helped us improve the quality of our manuscript. We have revised the manuscript accordingly.

1. Overall, I find the performance data shown in this work very strange. (a) First of all, Pt is very active for the reverse reaction (HOR and ORR). An amorphous overlayer such as chromium oxyhydroxide to cap the noble metal is critical for inorganic semiconductor-based photocatalysts to split water, as demonstrated by many studies of this field. In this regard, I have doubts that the Pt/CoO_x cocatalyst used in this work will absolutely cause reverse reactions. The authors checked ORR using the bare sample in Figure S12, which is meaningless. However, the authors did not perform reverse reaction examination over the Pt/CoO_x-modified photocatalyst, which should be critically checked. (b) Secondly, in the performance figures such as Figure S8, S14, S18a, there is a significant deactivation between adjacent runs; however, there is no deactivation within each run (the line is so straight within each run). This is impossible! (c) And in these cases, almost only hydrogen was produced, what were the oxidation products resulting from holes?

Response: The question is divided into three parts: (a), (b), and (c), each of which will be addressed separately.

Question (a)-1: Reverse reaction examination of the O₂ reduction at Pt site over the D-O-ZIS/Pt/CoO_x photocatalyst.

The O₂ reduction at Pt site in D-O-ZIS during photocatalytic process is lower than photocatalytic H₂ evolution, which is proven by theory and experimental results.

In theory, we investigated the re-reduction of O₂ at the Pt site using DFT calculations and energy band structure theory. The negative adsorption energy suggests that both H and O₂ are likely to be adsorbed on the Pt site, facilitating further redox reactions (**Figure R6a-c**). Additionally, the energy band structure of D-O-ZIS is depicted in **Figure R6d**. The conduction band (CB) of D-O-ZIS is negative enough for photocatalytic H₂ evolution (i.e., HER: H⁺+e⁻→1/2 H₂) and photocatalytic O₂ reduction (i.e., ORR: (1) e⁻+O₂→•O₂⁻; (2) •O₂⁻+e⁻+H₂O→OOH•+OH⁻; (3) OOH•+ H₂O+e⁻

$\rightarrow \text{H}_2\text{O}_2 + \text{OH}^-$, *Adv. Funct. Mater.* 2021, 31, 2105731). However, according to the Butler-Volmer equation (1)

$$j = j_0 \times \left\{ \exp \left[\frac{\alpha_a z F}{RT} (E - E_{eq}) \right] - \exp \left[- \frac{\alpha_c z F}{RT} (E - E_{eq}) \right] \right\} \quad (\text{R1})$$

where j is the current density, j_0 the exchange current density, E is the CB potential, E_{eq} is the equilibrium potential, T is the absolute temperature, z is the number of electrons involved in the photocatalytic reaction, F is the Faraday constant, R is the universal gas constant, α_a is the so-called cathodic charge transfer coefficient and α_c is the so-called anodic charge transfer coefficient. Compared with the ORR reaction, HER reaction possesses a higher potential difference between CB position and redox potential, which is favor for selectivity of HER during photocatalytic process. Therefore, both oxygen reduction reactions and hydrogen production reactions are feasible on Pt site.

Figure R6. Oxygen reduction reaction on D-O-ZIS/Pt/CoO_x. **a, b** Configuration of O_2 and H adsorption model on Pt site; **c** Adsorption energy of O_2 and H adsorption model on Pt site; **d** The energy band structure and redox potential of ZIS, D-ZIS and D-O-ZIS.

In experiment aspect, we conducted the isotopic measurement to evaluate the weight coefficient of HER and ORR on D-O-ZIS/Pt/CoO_x during the photocatalytic process. As shown in **Figure R7**, the photocatalytic H_2 evolution rate gradually decreases and stabilizes at around $1910 \mu\text{mol h}^{-1}$ when the amount of $^{18}\text{O}_2$ exceeded 2 mL. The consumed content of $^{18}\text{O}_2$ was 5.9%, which implies that the O_2 is slightly re-reduced on D-O-ZIS/Pt/CoO_x under conditions of competing hydrogen production reactions on Pt site.

Figure R7. Oxygen reduction reaction on D-O-ZIS/Pt/CoO_x. **a** Time-dependent photocatalytic overall water splitting profiles of D-O-ZIS/Pt/CoO_x under standard AM 1.5 illumination (100 mW cm⁻²), the mass of photocatalysts is 35 mg, Pt and CoO_x as cocatalyst, Na₂S/Na₂SO₃ acts as holes sacrificial agent; **b** Time-dependent photocatalytic water splitting profiles of D-O-ZIS/Pt/CoO_x by injecting different amount of ¹⁸O₂; **c** The rate of photocatalytic H₂ evolution rate of D-O-ZIS/Pt/CoO_x by injecting different amount of ¹⁸O₂.

In the revised version, **Figure R6 and R7** are added as a **Figure S15** in the supporting information; and the relative discussion has been added in the page 8 of main text and page 42 of supporting information and copied below:

The ¹⁸O₂ experiment showed(Figure S14). Furthermore, the D-O-ZIS with loaded cocatalysts (Pt and CoO_x) consumed 5.9% of the ¹⁸O₂ content, indicating the dominant role of Pt in the photocatalytic H₂ evolution process (Figure S15). (Page 8 of main text)

“Because the O₂ are prone to absorb on the Pt site (Figure S15a-c) and conduction band of D-O-ZIS is negative enough for photocatalytic O₂ reduction (Figure S15d) during the photocatalytic process, we conducted the isotopic measurement to evaluate the weight coefficient of photocatalytic H₂ evolution and photocatalytic O₂ reduction. As shown in Figure S15e, no O₂ gas evolution occurs with addition of Na₂S-Na₂SO₃ as sacrificial agent, indicating the photoinduced holes were consumed. Subsequently, we inject ¹⁸O₂ during photocatalytic process with addition of Na₂S-Na₂SO₃ as sacrificial agent to trace the O₂ reaction path. Along with the increase of ¹⁸O₂ amount, the photocatalytic H₂ evolution rate exhibits slight decrease and stabilizes at 1910 μmol h⁻¹ when the amount of ¹⁸O₂ exceeds 2 mL (Figure S15f, g). Meanwhile, 5.9% of the ¹⁸O₂ content was consumed in the D-O-ZIS with cocatalysts

(Pt and CoO_x) loading, implying the O₂ is hardly re-reduction on the Pt site.” (in the page 42 of supporting information)

Question (a)-2: Reverse reaction examination of the H₂ oxidation at Pt site over the Pt/CoO_x-modified photocatalyst.

The H₂ oxidation at Pt site in D-O-ZIS during photocatalytic process is lower than photocatalytic H₂ evolution, which is proven by theory and experimental results.

In theory aspect, we discuss the re-oxidation of H₂ at Pt site via DFT calculation and energy band structure theory. Firstly, we constructed H₂ or OH adsorption models on Pt and on CoO_x sites (**Figure R8a-c**). The corresponding adsorption energy was shown in **Figure R8d**. The negative adsorption energy indicates that the H₂ are more prone to adsorb on the Pt site with a more negative adsorption energy of -0.29 eV rather than on the CoO_x for further redox reaction. The OH adsorption on CoO_x is more conducive with a much negative adsorption energy of -0.36 eV. Meanwhile, the energy band structure of D-O-ZIS is shown in **Figure R6d**. The valence band (VB) of D-O-ZIS is positive enough for photocatalytic O₂ evolution (i.e., OER: 4OH+h⁺→2O₂) and H₂ oxidation (i.e., HOR: 2H₂+4OH⁻→4H₂O+4e⁻, *Nat. Commun.* 2021, 12, 2686). The OER reaction possesses a higher potential of redox potential, which is favor for selectivity of OER during photocatalytic process. Thus, the OER reaction is dominate reaction on catalyst during the photocatalytic process.

Figure R8. Hydrogen oxidation reaction on D-O-ZIS/Pt/CoO_x. a, b Configuration of H₂ and OH adsorption model on Pt site; c, d Configuration of H₂ and OH adsorption model on CoO_x site; e Adsorption energy of H₂ and OH adsorption model on Pt and CoO_x sites.

In experiment aspect, we conducted the isotopic measurement to evaluate the weight coefficient of HOR and OER on D-O-ZIS/Pt/CoO_x during the photocatalytic process. As shown in **Figure R9**, the photocatalytic O₂ evolution rate gradually decreases and stabilizes at around 523 μmol h⁻¹ when the amount of ²H₂ exceeded 3 mL. The consumed content of ²H₂ was 8.7%, which implies that the H₂ is hardly re-oxidized on D-O-ZIS, and it also has little influence on photocatalytic H₂/O₂ evolution.

Figure R9. Hydrogen oxidation reaction on D-O-ZIS/Pt/CoO_x. **a** Time-dependent photocatalytic overall water splitting profiles of D-O-ZIS/Pt/CoO_x under standard AM 1.5 illumination (100 mW cm⁻²), the mass of photocatalysts is 35 mg, Pt and CoO_x as cocatalyst, Na₂IO₃ acts as electrons sacrificial agent; **b** Time-dependent photocatalytic water splitting profiles of D-O-ZIS/Pt/CoO_x by injecting different amount of ²H₂; **c** The rate of photocatalytic O₂ evolution rate of D-O-ZIS/Pt/CoO_x by injecting different amount of ²H₂.

In the revised version, **Figure R8 and R9** are added as a **Figure S16** in the supporting information; and the relative discussion has been added in the page 8 of main text and page 43 of supporting information and copied below:

“The isotopic measurement further confirms that the H₂ re-oxidation reaction was suppressed over D-O-ZIS/Pt/CoO_x (Figure S16).” (Page 8 of main text)

“Because the H₂ are prone to absorb on the Pt site and CoO_x site (Figure S16a-d), and valence band of D-O-ZIS is positive enough for photocatalytic H₂ oxidation (Figure S16d) during the photocatalytic process, we conducted the isotopic measurement to evaluate the weight coefficient of photocatalytic O₂ evolution and photocatalytic H₂ oxidation. As shown in Figure S16e, no H₂ gas evolution occurs with addition of Na₂IO₃ as sacrificial agent, indicating the photoinduced electrons were consumed. Subsequently, we inject ²H₂ during photocatalytic process with

addition of Na_2IO_3 as sacrificial agent to trace the H_2 reaction path. Along with the increase of $^2\text{H}_2$ amount, the photocatalytic O_2 evolution rate exhibits slight decrease and stabilizes at $523 \mu\text{mol h}^{-1}$ when the amount of $^2\text{H}_2$ exceeds 3 mL (**Figure S16f, g**). Meanwhile, 8.7% of the $^2\text{H}_2$ content was consumed in the D-O-ZIS with cocatalysts (Pt and CoO_x) loading, implying the H_2 is hardly re-oxidation on the Pt and CoO_x sites.” (in the page 43 of supporting information)

Question (b): In the performance figures such as Figure S8, S14, S18a, when does the deactivation in catalytic performance occur?

We are trying to extend the testing time to observe the stability of the catalyst's performance. We extend the duration of each cycle to 20 hours to observe degrees of attenuation at each cycle. As shown in **Figure R10**, ZIS and D-ZIS with cocatalysts loading exhibit attenuation of photocatalytic activity when extending the cycle time to 20 h. ZIS and D-ZIS exhibit significant attenuation in the absence of cocatalysts during the reaction as shown in **Figure R11**. Thus, after a prolonged reaction time, the catalyst structure of ZIS and D-ZIS becomes deactivated and unstable, resulting in decreased and poor photocatalytic performance.

In the revised version, **Figure R10 and R11** are added as a new **Figure S9** and **Figure S18** in the supporting information; and the relative discussion has been added in the page 8 of main text and page 30 of supporting information and copied below:

*“ZIS and D-ZIS cannot split water into H_2 and O_2 with co-catalyst loading and activity attenuation occurs in each cycle (**Figure S9**). (Page 8 of main text)*

*“In contrast, ZIS and D-ZIS only produced H_2 and showed a decrease in catalytic activity in each cycle (**Figure S18**). (Page 8 of main text)*

*“We extend the duration of each cycle to 20 hours to observe degrees of attenuation at each cycle. As shown in **Figure S9c, d**, ZIS and D-ZIS with cocatalysts loading exhibit attenuation of photocatalytic activity when extending the cycle time to 20 h. Thus, after a prolonged reaction time, the catalyst structure of ZIS and D-ZIS becomes deactivated and unstable, resulting in decreased and poor photocatalytic performance.” (Page 30 of supporting information)*

Figure R10. Photocatalytic overall water splitting performance. Time-dependent photocatalytic overall water splitting over **a** ZIS and **b** D-ZIS, in pure water under standard AM 1.5 illumination (100 mW cm^{-2}), Pt and CoO_x used as cocatalysts, Pt to CoO_x wt% ratio of 1:4, the photocatalyst mass was 35 mg and the photocatalytic activity was evaluated via the total hydrogen and oxygen yield of a cycle, the time of each cycle is 20 h.

Figure R11. The overall water splitting performance of ZIS and D-ZIS without cocatalysts loading. Time-dependent photocatalytic overall water splitting over **a** ZIS and **b** D-ZIS in pure water. Reaction conditions: under standard AM1.5G illumination (100 mW cm^{-2}), the photocatalyst mass is 35 mg and the photocatalytic activity is evaluated via the total hydrogen and oxygen yield of a cycle, the time of each cycle is 20 h.

Question (c): What were the oxidation products resulting from holes?

Response: When sacrificial hole scavengers are not added, sulfides undergo photocatalytic water decomposition reactions, during which lattice S^{2-} ions are more easily escaped and oxidized by photoexcited holes into SO_4^{2-} , thus leading to deactivation of photocatalysts. The photo-corrosion of photocatalysts for ZIS and D-ZIS induced by S leaching after the photocatalytic reaction was

confirmed by S 2p XPS (i.e. **Figure 3e**). The S 2p spectra of ZIS and D-ZIS showed lower binding energy shifts of 0.62 and 0.59 eV, respectively, with reduced intensities, indicating that S leaching in ZIS and D-ZIS occurred and deactivating the catalyst. We confirmed the presence of a small peak corresponding to SO_4^{2-} in the S 2p spectrum (**Figure R12a**) after the reaction. The graph has been updated accordingly in the main text. The binding energy peak of SO_4^{2-} at 169.89 eV is observed in both ZIS and D-ZIS after the reaction (*Adv. Energy Mater.* 2021, 11, 2101181). This indicates that oxygen, present during the water splitting process of ZIS and D-ZIS, disrupts the atomic structure of S, leading to the formation of adsorbed SO_4^{2-} ions on the material surface. Additionally, the HRTEM image of ZIS and D-ZIS after testing revealed S leaching, resulting in the formation of a S-hole structure and a compromised shell thickness (**Figure R12b, c**).

Figure R12. Photocatalytic performance. **a** The S 2p XPS spectra of ZIS and D-ZIS before and after 120 h photocatalytic test. The vertical bars indicate the difference in intensity before and after test; HRTEM of ZIS and D-ZIS after photocatalytic testing. **b** ZIS, **c** D-ZIS.

In the revised version, **Figure R12a** has been revised in **Figure 3e** in the main text. **Figure R12b, c** are added as new **Figure S22** in the supporting information; and the relative discussion has been revised in the page 9 of main text, page 51 of supporting information, and copied below:

“The S 2p spectra of D-O-ZIS showed with reduced intensities and oxidation product of SO_4^{2-} , indicating that S leaching..... The HRTEM image and structural analysis of D-O-ZIS....., while ZIS and D-ZIS exhibited S leaching characteristics (Figure S22)”. (Page 9 of main text)

“ZIS and D-ZIS suffer from corrosion under light irradiation, during which lattice S^{2-} ions are more easily

escaped and oxidized by photoexcited holes into SO_4^{2-} , thus leading to deactivation of photocatalysts. The HRTEM image of ZIS and D-ZIS after testing revealed S leaching features, forming S atom loss structures and the shell thickness for D-ZIS is destroyed.” (Page 51 of supporting information)

2. Cocatalysts decoration methods should have been given in detail, as they are very important components for a photocatalyst. The structure, dispersion and chemical states of each cocatalyst component should be analyzed. However, these data are unexpectedly missing. And why was the activity almost the same by loading significantly different ratios of Pt/CoO_x (1:4 and 3:1) in Figure S7a and b?

Response: Thanks very much for the valuable comments. We have supplemented the preparation method of the cocatalysts loading, the microstructure, and elemental composition and chemical states of cocatalysts to further study the properties of cocatalysts (Pt and CoO_x). D-O-ZIS demonstrates distinct hydrogen production rates of 31.09 and 25.55 $\mu\text{mol h}^{-1}$, respectively, with Pt/CoO_x loading ratios of 1:4 and 3:1. Increasing Pt or CoO_x loading initially improves performance, providing more active sites and promoting charge separation and transfer. However, excessive Pt or CoO_x content hinders the catalytic sites and caused a shielding effect on the photocatalyst, resulting in a decline in performance (*Chem. Soc. Rev.*, 2014, 43, 7787).

Preparation method of the cocatalysts loading

Pt cocatalyst loading. In a typical procedure of photochemical loading Pt on D-O-ZIS photocatalyst, 35 mg D-O-ZIS photocatalyst and H₂PtCl₆ (1.5 mg mL⁻¹) were dispersed in an aqueous solution containing 50 mL H₂O. After the light irradiation under standard AM1.5G illumination (100 mW cm⁻²) for 1 h, the photocatalyst was centrifuged and washed by deionized water and then dried, yielding D-O-ZIS/Pt catalyst.

CoO_x cocatalyst preparation and loading. 0.5 g of Co(NO₃)₂ was dissolved in 20 mL of deionized water and stirred. The solution was heated on a heating plate until complete evaporation of water. Pink precipitate was observed during the process and the resulting precipitate was then subjected to thermal decomposition in a muffle furnace at 400 °C for 2 h. Following the thermal treatment, black particles were obtained and designated as CoO_x.

Taking a Pt:CoO_x loading weight ratio (wt%) of 1:4 as an example, 35 mg of catalyst D-O-ZIS was dispersed in 50 mL of deionized water. Then, 2.1 mg of CoO_x was added, and the dispersion was

sonicated for 30 minutes, resulting in the D-O-ZIS/CoO_x catalyst. Subsequently, a solution of 1.5 mg mL⁻¹, 0.35 mL of H₂PtCl₆ was added for photodeposition reaction. The mixture was irradiated under light for 1 h, yielding D-O-ZIS/Pt/CoO_x catalyst.

Figure R13. Cocatalyst characterization. **a** HRTEM image of D-O-ZIS/Pt/CoO_x presenting both Pt nanoparticles and CoO_x nanosheets loading on D-O-ZIS/Pt/CoO_x; **b** XPS spectra of Pt 4f in D-O-ZIS/Pt/CoO_x; **c** XPS spectra of Co 2p in D-O-ZIS/Pt/CoO_x; **d** XPS spectra of O 1s in D-O-ZIS/Pt/CoO_x.

Microstructure analysis of D-O-ZIS/Pt/CoO_x. The detailed microstructure of D-O-ZIS/Pt/CoO_x was further studied by the high-resolution TEM image, as shown in **Figure R13a**. As indicated, the D-O-ZIS/Pt/CoO_x presents the decoration of Pt nanoparticles with ~15 nm and CoO_x nanosheet of 20 nm. The resolved lattice distances of 0.23 nm match well with the (111) plane of Pt, while a lattice distance of 0.24 nm is attributed to the (311) plane of CoO_x (*Angew. Chem. Int. Ed.* 2022, 61, e202116057).

Elemental composition and chemical states of cocatalysts. The X-ray photo-electron spectroscopy (XPS) of Pt 4f spectrum of D-O-ZIS/Pt/CoO_x presents two dominating peaks located at 69.0 and 72.3 eV (**Figure R13b**), which belongs to metallic Pt (*J. Mater. Chem. A*, 2015, 3, 17154). The other two peaks at 69.9 and 73.4 eV are attributed to Pt²⁺, which is probably owing to the constructed Pt single atoms (*Nat. Commun.*, 2022, 13, 1287). As shown in **Figure R13c**, the Co 2p spectrum of D-O-ZIS/Pt/CoO_x showed the binding energy of Co³⁺ peak (775.9 and 791.6 eV) and Co²⁺ peak (777.6 and 793.9 eV), indicating the formation of the CoO_x structure (*Angew. Chem. Int. Ed.* 2022, 61,

e202116057). As shown in **Figure R13d**, the deconvoluted O 1s spectrum of D-O-ZIS/Pt/CoO_x exhibits two peaks located at 530.7 and 531.8 eV representing the lattice oxygen (CoO) and oxygen deficiency, respectively (*Angew. Chem. Int. Ed.* 2022, 61, e202116057). The other three peaks are attributed to the OH vibration, H₂O peak, and S-O bonding peak.

In the revised version, **Figure R13** is added as a new **Figure S8** in the supporting information; and the relative discussion has been added in the page 6 and 27 of supporting information and copied below:

“Preparation method of the cocatalysts loading.....” (Page 6 of supporting information)

*“.....The detailed microstructure of D-O-ZIS/Pt/CoO_x was further studied by the high-resolution TEM image, as shown in **Figure S8a**. D-O-ZIS/Pt/CoO_x presents the decoration of Pt nanoparticles with ~15 nm and CoO_x nanosheet of 20 nm. The resolved lattice distances of 0.23 nm match well with the (111) plane of Pt, while a lattice distance of 0.24 nm is attributed to the (311) plane of CoO_x^[25, 26]. As shown in **Figure S8b**, the XPS of Pt 4f spectrum of D-O-ZIS/Pt/CoO_x presents two dominating peaks located at 69.0 and 72.3 eV, which belongs to metallic Pt. The other two peaks at 69.9 and 73.4 eV are attributed to Pt²⁺, which is probably owing to the constructed Pt single atoms^[20]. As shown in **Figure S8c**, the Co 2p spectrum of D-O-ZIS/Pt/CoO_x showed the binding energy of Co³⁺ peak (775.9 and 791.6 eV) and Co²⁺ peak (777.6 and 793.9 eV), indicating the formation of the CoO_x structure^[25, 26]. As shown in **Figure S8d**, the deconvoluted O 1s spectrum of D-O-ZIS/Pt/CoO_x exhibits two peaks located at 530.7 and 531.8 eV representing the lattice oxygen (CoO) and oxygen deficiency, respectively. The other three peaks are attributed to the OH vibration, H₂O peak, and S-O bonding peak.....” (Page 27 of supporting information)*

3. The weight-dependent test showed a saturated rather than linear increased activity with the increased amount of photocatalyst (**Figure S9b**). Therefore, it is unreasonable to normalize the activity with gram in **Figure 3d** and **Figure S7c**.

Response: Thanks very much for the valuable comments. We have revised the unit of photocatalytic activity evaluation. As shown in **Figure R14**, the production rate of H₂ and O₂ is evaluated as the photocatalytic activity based on the calculation in $\mu\text{mol h}^{-1}$.

In the revised version, **Figure R14a, b, c, d** is added as a new **Figure 3d, S8g, S19b, d**, respectively, in the main text and supporting information.

Figure R14. Photocatalytic performance. **a** Photocatalytic gas yield of ZIS, D-ZIS, and D-O-ZIS before and after photocatalytic overall water splitting test in pure water. Error bars represent the standard deviations from the statistic results of three sets of experiments; **b** Photocatalytic gas evolution rate of D-O-ZIS and D-O-ZIS/Pt/CoO_x with different Pt:CoO_x ratios for photocatalytic overall water splitting test in pure water; **c** Time-dependent photocatalytic O₂ evolution half-reaction for ZIS, D-ZIS and D-O-ZIS in the presence of sacrificial reagents (20 mM NaIO₃ aqueous solution); **d** Photocatalytic O₂ gas evolution rate of D-O-ZIS before and after photocatalytic tests. Under standard AM 1.5 illumination (100 mW cm^{-2}), the photocatalyst mass is 35 mg and the photocatalytic activity is evaluated via the total oxygen yield of a cycle, the time of each cycle is 12 h. Error bars represent the standard deviations from the statistic results of three sets of experiments.

4. According to the equation of $\text{IO}_3^- + 6e^- + 3\text{H}_2\text{O} = \text{I}^- + 6\text{OH}^-$, 50 mL of 20 mM NaIO₃ aqueous contains 1 mmol NaIO₃, which consumes 6 mmol of electron and therefore 1.5 mmol of O₂ can be evolved. Therefore, it's impossible to produce ~1.6 mmol O₂ in the first 12 hours and continue to consume electrons after 12 hours in Figure S15c.

Response: Thank you for your reminder. In the process of the oxygen production half-reaction, a Na₂IO₃ solution is introduced into the reactor as an electron sacrificial agent. We conducted multiple tests on oxygen evolution in the testing data for O₂ and have recalibrated the standard curve for the photocatalytic water decomposition injection system. We have made a re-evaluation of the oxygen production value based on the revised standard curve for O₂ testing data.

To create a new standard curve, a manual sampling system was utilized. Oxygen was injected at regular 30-minute intervals in a fixed volume of 0.5 mL, allowing the sample to pass through the chromatographic column. The peak area values obtained from the chromatographic column at each standard concentration of O₂ were recorded at these six data collection points. These sampling data

were then utilized to construct the standard curve. A new standard curve was derived from the sampling data (**Figure R15a**), yielding the peak area value for a 1 mL oxygen injection. Consequently, injecting 1 mL of oxygen into the chromatographic column produced an oxygen peak area of 12625.08356. The measured oxygen amount has been re-evaluated and revised in **Figure R15b**. After conducting multiple experiments, error bars were added to the data. The revised oxygen yield is determined to be 1454.2 μmol , which is lower than the corresponding electron consumption of the added sacrificial agent (1500 μmol). A cycle is carried out every 12 hours, followed by replenishing the concentration of the sacrificial agent, and then evacuation is continued for testing purposes.

Figure R15. Photocatalytic performance. **a** The standard curve for O_2 testing of peak area obtained from the chromatographic column for each standard concentration of O_2 ; Error bars represent the standard deviations from the statistic results of three sets of experiments; **b** Time-dependent photocatalytic O_2 evolution half-reaction for ZIS, D-ZIS and D-O-ZIS in the presence of sacrificial reagents (20 mM NaIO_3 aqueous solution). Error bars represent the standard deviations from the statistic results of three sets of experiments.

In the revised version, the O_2 production amount in **Figure 3a, d, S8e-g, S10b, S17a, S19c, d, S24, Table S4, S8, S12** have been revised according to the new standard curve for O_2 testing in the main text and supporting information.

5. Some characterization data are not correctly interpreted. For example, the EPR signal at g value of 2.004 was assigned by the authors to zinc vacancies. However, this signal, to be correct, should be assigned to free electrons or unpaired electrons trapped in a vacancy. Individual zinc vacancies will not produce an EPR signal, and in my opinion, the detected EPR signal at g value of 2.004 probably results from unpaired electrons trapped in anion (O or S) vacancies. ICP is suggested to study the loss of zinc.

Response: Thanks very much for the valuable comments. Electron spin resonance (ESR) spectroscopy

of D-ZIS displayed a peak intensity at $g = 2.004$ attributed to the Zn vacancy, which has been corroborated by several literature as well (*Adv. Funct. Mater.* 2019, 29, 1905153; *J. Am. Chem. Soc.* 2017, 139, 7586). The ESR signal at a g -factor of 2.009 confirms the S-vacancies (*Nat. Commun.* 2021, 12, 4112).

Additionally, inductively coupled plasma (ICP) emission spectrometer instrument was employed to accurately analyze the amounts of Zn, In and S in ZIS, D-ZIS, and D-O-ZIS. The atomic ratio of Zn, In and S in ZIS, D-ZIS, and D-O-ZIS is 1: 2.05 :3.93, 0.91:2.05:3.92, and 0.86:2.05:3.93, respectively. From the comparison about the atomic proportions of ZIS, D-ZIS, and D-O-ZIS, it can be identified that the Zn vacancies exist in D-ZIS and D-O-ZIS. ICP analysis revealed that the Zn vacancy content in D-ZIS is approximately 1.01 wt%, while in D-O-ZIS, it is approximately 2.06 wt%.

Table R1. ICP elemental analysis of ZIS, D-ZIS, and D-O-ZIS

	Zn (mg L ⁻¹)	In (mg L ⁻¹)	S (mg L ⁻¹)
ZIS	3.67	12.31	7.75
D-ZIS	3.43	12.32	7.71
D-O-ZIS	3.18	12.31	7.72

In the revised version, **Table R1** was added as a new **Table S2** in the supporting information; and the relative discussion has been added in the page 6 of main text and page 21 of supporting information and copied below:

“Electron spin resonance (ESR).....(Figure S3c, Table S1), which is consistent with the inductively coupled plasma (ICP) emission spectrometer results (Table S2).” (Page 6 of main text)

“Inductively coupled plasma (ICP) emission spectrometer instrument was employed to accurately analyze the amounts of Zn, In and S in ZIS, D-ZIS, and D-O-ZIS. The atomic ratio of Zn, In and S in ZIS, D-ZIS, and D-O-ZIS is 1: 2.05 :3.93, 0.91:2.05:3.92, and 0.86:2.05:3.93, respectively. From the comparison about the atomic proportions of ZIS, D-ZIS, and D-O-ZIS, it can be identified that the Zn vacancies exist in D-ZIS and D-O-ZIS. ICP analysis revealed that the Zn vacancy content in D-ZIS is approximately 1.01 wt%, while in D-O-ZIS, it is approximately 2.06 wt%.” (Page 21 of supporting information)

Reviewer #3 (Remarks to the Author):

*This manuscript proposes an electronegativity difference strategy to activate and stabilize ZIS for photocatalytic overall water-splitting, achieving a remarkable 0.57% solar-to-hydrogen conversion efficiency along with high stability. A distortion-evoked cation-site O doping in Zn atom sites of D-O-ZIS generates significant electronegativity differences between adjacent atomic sites, with S_1 sites being electron-rich and S_2 sites being electron-deficient in the local S_1 - S_2 -O structure. The strong charge redistribution character activates stable oxygen reactions at S_2 sites and hydrogen adsorption/desorption at S_1 sites. **This is a carefully done study and the findings are of considerable interest. A few minor revision** are list below.*

Response: We thank the referee for appreciating the impact and value of our study. We also appreciate the referee's constructive comments, which have helped us improve the quality of our manuscript. We have revised the manuscript accordingly.

1. How to determine the content of zinc vacancies by Zn XPS spectra (Figure S3c)? The author needs to provide the depth explanation.

Response: Thanks very much for the valuable comments. The detailed information regarding the XPS fits for the determination of Zn vacancies is listed in **Table R2**. The Zn content can be approximately calculated based on the XPS peak area of Zn 2p. By comparing the Zn element content of ZIS, D-ZIS, and D-O-ZIS, the approximate amount of Zn loss can be determined. The lost Zn content in D-ZIS is estimated to be 2.3%, whereas in D-O-ZIS, it is approximately 5.8%.

Table R2. Zn 2p XPS fitting data of ZIS, D-ZIS, and D-O-ZIS

Catalysts	Zn 2p _{3/2}			Zn 2p _{1/2}		
	Peak (eV)	FWHM	Area	Peak (eV)	FWHM	Area
ZIS	1022.7	1.62	6163.4	1045.6	1.72	4003.9
D-ZIS	1022.9	1.61	5987.9	1045.9	1.70	3946.5
D-O-ZIS	1023.1	1.63	5893.8	1046.2	1.73	3681.2

In the revised version, **Table R2** was added as a new **Table S1** in the supporting information; and the relative discussion has been added in the page 6 of main text and page 21 of supporting information and copied below:

“Electron spin resonance (ESR)....., with a concentration of ~2.3% determined by Zn XPS spectra (Figure S3c, Table S1),” (Page 6 of main text)

“The Zn content can be approximately calculated based on the XPS peak area of Zn 2p. By comparing the Zn element content of ZIS, D-ZIS, and D-O-ZIS, the approximate amount of Zn loss can be determined. The lost Zn content in D-ZIS is estimated to be 2.3%, whereas in D-O-ZIS, it is approximately 5.8%.” (Page 21 of supporting information)

2. Why does the incorporation of O atoms lead to the formation of more Zn vacancies? To further prove it, it is recommended that the author provide comparative experimental results.

Response: Thanks very much for the comments. We utilized DFT calculations to investigate the impact of different O doping concentrations on the structure of D-ZIS as shown in **Figure R16**. Specifically, we examined the effects of one, two, and three oxygen atoms being doped. Our findings reveal that when one oxygen atom is doped, it has a propensity to occupy the position of a Zn atom in the resulting structure, leading to the creation of a neighboring Zn vacancy. This phenomenon can induce structural distortion. Moreover, the introduction of two or three oxygen atoms further promotes the formation of Zn vacancies, thereby intensifying the degree of structural distortion. The doping of oxygen atoms induces a reduction in the formation energy of Zn vacancies. This can be attributed to the dissimilar atomic sizes and chemical properties of oxygen and Zn atoms, which generate stress and distortion during the doping process. Consequently, these distortions facilitate the formation of Zn vacancies as the structure accommodates the newly introduced oxygen atoms.

In the revised version, **Figure R16** was added as new **Figure S6**. The relative discussion has been added in the page 7 of main text and page 25 of supporting information and copied below:

“....., indicating an increase in distortion states in the structure (Figure 2d). This is consistent with the formation of Zn vacancies induced by O doping in theoretical calculations (Figure S6).” (Page 7 of main text)

“We utilized DFT calculations to investigate the impact of different O doping concentrations on the structure of D-ZIS as shown in Figure S6. Specifically, we examined the effects of one, two, and three O atoms being doped. Our findings reveal that when one O atom is doped, it has a propensity to occupy the position of a Zn atom in the resulting structure, leading to the creation of a neighboring Zn vacancy. This phenomenon can induce structural distortion. Moreover, the introduction of two or three O atoms further promotes the formation of Zn vacancies, thereby intensifying the degree of structural distortion.” (Page 25 of supporting information)

Figure R16. Structural transition models and formation energies at different O doped concentrations.

3. There is a lack of comparison for AQY and STH of D-O-ZIS and other $ZnIn_2S_4$ based nanocomposites for overall water splitting. The author should add it and discuss in more detail.

Response: Thanks very much for the valuable comments. We have included a comparison of the AQY and STH efficiency of D-O-ZIS and other $ZnIn_2S_4$ -based nanocomposites for overall water splitting as shown in **Table R3** and **Figure R17**. The photocatalytic performance of D-O-ZIS surpasses that of the majority of composite heterogeneous photocatalysts, demonstrating significant advantages in the field of photocatalysis.

Table R3. Comparison of photocatalytic overall water splitting in reported $ZnIn_2S_4$ -based nanocomposites photocatalysts.

Catalysts	Reaction conditions	H ₂ evolution rate ($\mu\text{mol h}^{-1}$)	O ₂ evolution rate ($\mu\text{mol h}^{-1}$)	AQY (%)	STH (%)	Ref.
D-O-ZIS	300 W Xe lamp, 0.035 g catalyst, 2wt% Pt and 8wt% CoO _x as cocatalysts	31.1 ($\lambda > 300$ nm)	13.7 ($\lambda > 300$ nm)	14.90@400 nm	0.57	This work
Nb ₄ C ₃ T _x	300 W Xe lamp,	1.07	0.53	1.2@380 nm	0.021	[R1]
Mxene@ZnIn ₂ S ₄	0.02 g catalyst,	($\lambda > 420$ nm)	($\lambda > 420$ nm)			
PtS-ZnIn ₂ S ₄ /WO ₃ -	300 W Xe lamp,	0.74	0.28	0.50@420 nm	0.013	[R2]

MnO ₂	0.05 g catalyst, 0.5% PtS and 3.0% MnO ₂ as cocatalyst.	(λ>420 nm)	(λ>420 nm)			
NiCo ₂ S ₄ /ZIS/Co ₃ O ₄	300 W Xe lamp, 0.1 g catalyst, Pt, and Co ₃ O ₄ as cocatalyst.	1.06 (λ>400 nm)	0.28 (λ>400 nm)	13.5@400 nm	0.020	[R3]
TiO ₂ -ZnIn ₂ S ₄	300 W Xe lamp, 0.02 g catalyst	4.29 (λ>420 nm)	1.63 (λ>420 nm)	36.17@365 nm	0.078	[R4]
ZnIn ₂ S ₄ /RGO/Bi ₂ Mo ₆	300 W Xe lamp, 0.1 g catalyst, Pt and CoO _x as cocatalyst.	3.08 (λ>420 nm)	1.56 (λ>420 nm)	/	0.057	[R5]
ZnIn ₂ S ₄ -Au-TiO ₂	300 W Xe lamp, 0.05 g catalyst	9.32 (λ>420 nm)	3.31 (λ>420 nm)	/	0.17	[R6]
BiVO ₄ @ZnIn ₂ S ₄ /Ti ₃ C ₂	300 W Xe lamp, 0.06 g catalyst	6.16 (λ>400 nm)	3.05 (λ>400 nm)	2.4@410 nm	0.12	[R7]
BiOBr/ZnIn ₂ S ₄	300 W Xe lamp, 0.1 g catalyst, Pt as cocatalyst.	62.8 (λ>420 nm)	30.4 (λ>420 nm)	8.57@420 nm	1.14	[R8]
BiFeO ₃ /ZnIn ₂ S ₄	300 W Xe lamp, 0.012 g catalyst.	0.77 (λ>420 nm)	0.38 (λ>420 nm)	24.28@365 nm	0.014	[R9]
InVO ₄ @ZnIn ₂ S ₄	300 W Xe lamp, 0.005 g catalyst.	7.86 (λ>420 nm)	3.06 (λ>420 nm)	0.34@450 nm	0.140	[R10]
Sulfur-Defficient ZnIn ₂ S ₄ /Oxygen- Defficient WO ₃	300 W Xe lamp, 0.035 g catalyst. Pt and CoO _x as cocatalyst.	169.2 (λ>420 nm)	86.5 (λ>420 nm)	3.4@380 nm	1.52	[R11]

Figure R17. Comparison of photolytic overall water splitting for the ZnIn₂S₄-based nanocomposites photocatalysts. (see **Table R3** for details). We calculated STH values using the provided equation for references that contained relevant parameters. For literature that did not offer relevant parameters, we could not calculate the STH values.

In the revised version, **Table R3** was added as a new **Table S10** in the supporting information; and **Figure R17** was added as new **Figure S13**. The relative discussion has been added in the page 8 of main text and copied below:

“The solar to hydrogen (STH) efficiency..... , which outperforms most of the recently reported single photocatalysts (Figure S12, Table S9) and composite heterogeneous photocatalysts (Figure S13, Table S10).” (Page 8 of main text)

4. The photolytic H₂ and O₂ evolution amounts up to 373.2 and 178.8 μmol, the ratio of H₂ and O₂ exceeds 2:1, why is O₂ evolution amounts so high ?

Response: Thanks very much for the valuable comments. The photolytic water splitting reaction resulted in the evolution of 373.2 μmol of H₂ and 178.8 μmol of O₂, with a hydrogen-to-oxygen production ratio of approximately 2.09. To validate this ratio, we performed multiple experiments with error bars added to the obtained data. The experiments yielded a hydrogen-to-oxygen gas production ratio of approximately 2.18. The revised hydrogen and oxygen yield is determined to be 373 and 171 μmol as shown in **Figure R18**.

Figure R18. Time-dependent photocatalytic overall water splitting over D-O-ZIS in pure water under standard AM 1.5 illumination (100 mW cm^{-2}), Pt and CoO_x used as cocatalysts, Pt to CoO_x wt% ratio of 1:4, the photocatalyst mass was 35 mg and the photocatalytic activity was evaluated via the total hydrogen and oxygen yield of a cycle, the time of each cycle is 12 h. Error bars represent the standard deviations from the statistic results of three sets of experiments.

In the revised version, the O_2 production amount and correspondong evolution rate has been revised in **Figure 3a** and **Figure 3b** in the main text.

5. In order to confirm the stability of the sample, the author is requested to provide the XRD, XPS, SEM, TEM and HRTEM images of the sample after the photocatalytic reaction.

Response: Thanks very much for the valuable comments. We have added XRD and XPS analysis, SEM, TEM, and HRTEM images of the sample after photocatalytic testing.

As shown in **Figure R19**, after the photocatalytic reaction of D-O-ZIS, the catalyst maintains its flower-like layered structure (**Figure R19a, b**). High-resolution transmission electron microscopy (HRTEM) analysis reveals that the lattice spacing of D-O-ZIS remains at 0.32 nm, and the atomic lattice arrangement within the planes remains ordered, while the edges exhibit a distorted structure (**Figure R19c, d**). The intensity of XRD peaks for D-O-ZIS remains almost unchanged before and after the reaction, with negligible shifts, indicating the preservation of the crystal structure (**Figure R19e**). Furthermore, the XPS spectra of Zn 2p and In 3d demonstrate minimal changes before and after the photocatalytic reaction (**Figure R19f, g**). This results further indicate its excellent structural stability for photocatalytic reactions.

In the revised version, **Figure R19** was added as a new **Figure S21** in the supporting information. The relative discussion has been added in the page 9 of main text and page 50 of supporting information and copied below:

*“The HRTEM image and structural analysis of D-O-ZIS after testing revealed stable distortion features (**Figure 3f, S21**).....” (Page 9 of main text)*

“As shown in **Figure S21**, after the photocatalytic reaction of D-O-ZIS, the catalyst maintains its flower-like layered structure (**Figure S21a, b**). High-resolution transmission electron microscopy (HRTEM) analysis reveals that the lattice spacing of D-O-ZIS remains at 0.32 nm, and the atomic lattice arrangement within the planes remains ordered, while the edges exhibit a distorted structure (**Figure S21c, d**). The intensity of XRD peaks for D-O-ZIS remains almost unchanged before and after the reaction, with negligible shifts, indicating the preservation of the crystal structure (**Figure S21e**). Furthermore, the XPS spectra of Zn 2p and In 3d demonstrate minimal changes before and after the photocatalytic reaction (**Figure S21f, g**).” (Page 50 of supporting information)

Figure R19. Structural stability of the D-O-ZIS after 120 h photocatalytic test. **a** SEM image of D-O-ZIS after photocatalytic testing; **b** TEM image of D-O-ZIS after photocatalytic testing; **c, d** HRTEM images of D-O-ZIS after photocatalytic testing; **e** XRD patterns of D-O-ZIS before and after photocatalytic testing; **f** XPS spectra of Zn 2p in D-O-ZIS before and after photocatalytic testing; **g** XPS spectra of In 3d in D-O-ZIS before and after photocatalytic testing.

REVIEWER COMMENTS

Reviewer #1 (Remarks to the Author):

Thanks to Li and Xu et al. for their thoughtful responses to my previous questions. Overall, the answers were quite rational and satisfying. Therefore, I suggest its publication on Nat Commun.

Reviewer #2 (Remarks to the Author):

Due to the critical issues below, I question the accuracy of the data, and would reject this paper.

1. I'm surprised that D-O-ZIS/Pt/CoOx can produce ~24 mmol H₂ within 12 h under AM1.5G irradiation without obvious activity degradation (Figure R7a). What is the volume of the empty space of the reaction system, and how is the accuracy (and applicable range) of the calibration curve for the H₂ gas? As far as we know, the volume of the empty space for the Labsolar 6A loop (manufactured by Beijing Perfectlight Technology Co., Ltd.) is about 130 mL. In this case, the pressure generated by 24 mmol H₂ is estimated to be 457 kPa. I can't believe the glass reaction system can work under such high inside pressure, especially the valves sealed by grease.

2. The amount of O₂ for the reverse reaction test is too small compared with the produced H₂ in Figure R7b. For overall water splitting reactions, the produced amount of H₂ and O₂ are comparable (2:1 in molar ratio). Therefore, it would make sense to introduce a comparable amount of O₂ for the reverse reaction examination, for example, to add around 12 mmol O₂ in Figure R7b. It is not necessary to use isotopic ¹⁸O₂; quantification of the change in the amount of H₂ and O₂ by gas chromatography is enough to address the issue. The same for Figure R9b, in which ~12 mmol H₂ should be introduced. In addition, it is incorrect to use the term "photocatalytic overall water splitting" in the captions of Figure R7 and R9 because these are sacrificial H₂ and O₂ evolution reactions (NOT overall water splitting).

3. How about the reverse reaction over D-O-ZIS/Pt/CoOx under dark conditions? That is, introduce a certain amount of H₂ and O₂ with a molar ratio of 2:1 (for example, 4 mmol H₂ and 2 mmol O₂) to the reaction system and check the time-dependent change in the amount of H₂ and O₂ without light irradiation. This simple experiment is what we initially suggest the authors do, for the sake of figuring out the reverse reactions over the bare Pt on the surface.

4. The authors show the standard curve for O₂ in Figure R15a, in which 3 mL (equivalent to around 134 μmol) O₂ at maximum was used for the calibration. Considering that the response coefficient of TCD is not linearly proportional to the amount of O₂, the applicable range of this standard curve should be limited to around 134 μmol. However, the tested O₂ amount was beyond 1200 μmol. This makes me doubt again the accuracy of the data in this work.

5. The comment of “why was the activity almost the same by loading significantly different ratios of Pt/CoOx (1:4 and 3:1) in Figure S7a and b” is not well addressed. Is that a coincidence or is there any underlying reason?

6. The deposition method of CoOx in this work is very strange. What is the particle size and morphology of the as-prepared CoOx particles? I do not believe that just mixing the CoOx powder and D-O-ZIS in water would form CoOx-decorated D-O-ZIS. In my opinion, this step will form segregated CoOx and D-O-ZIS, without intimate contact between them. A widely used method for unselectively depositing the CoOx cocatalyst is impregnation with Co(NO₃)₂ (or other cobalt precursors) followed by thermal treatment. Did the authors try the impregnation method?

7. HRTEM is not reliable to analyze the CoOx and Pt species on D-O-ZIS. The HRTEM image (Figure S8a) is unduly treated, making it hard to recognize the real surface structure (and the original HRTEM image). STEM-EDS should be performed to clearly understand whether or not CoOx was deposited (see the comment above) and the nanostructures of Pt (and CoOx) formed.

8. EPR is a technique for detecting unpaired electrons. A vacancy itself will not produce an EPR signal. When (unpaired) free electrons were trapped in specific vacancies, it may generate an EPR signal at g value of around 2.0023. Therefore, the expressions of “...g = 2.004 attributed to the Zn vacancy” and similar were not scientifically correct. According to the authors’ response, I do not think that they understand EPR and this issue was not addressed in the revised manuscript.

Reviewer #3 (Remarks to the Author):

The authors have addressed all my comments. Therefore, I'd like to recommend its publication in the present form now.

Point-by-point Response to the Reviewers' Comments

Reviewer #1 (Remarks to the Author):

Thanks to Li and Xu et al. for their thoughtful responses to my previous questions. Overall, the answers were quite rational and satisfying. Therefore, I suggest its publication on Nat Commun.

Response: Thanks for your comment and recognition.

Reviewer #3 (Remarks to the Author):

The authors have addressed all my comments. Therefore, I'd like to recommend its publication in the present form now.

Response: Thanks for your comment and recognition.

Reviewer #2 (Remarks to the Author):

Due to the critical issues below, I question the accuracy of the data, and would reject this paper.

Response: In response to the questions raised by the reviewer, we have conducted further refinement and enhancement of the testing and experimental detail. We have revised the manuscript accordingly. Below are the detailed responses.

1. **(a)** I'm surprised that D-O-ZIS/Pt/CoO_x can produce ~24 mmol H₂ within 12 h under AM1.5G irradiation without obvious activity degradation (Figure R7a). **(b)** What is the volume of the empty space of the reaction system, as far as we know, the volume of the empty space for the Labsolar 6A loop (manufactured by Beijing Perfectlight Technology Co., Ltd.) is about 130 mL. In this case, the pressure generated by 24 mmol H₂ is estimated to be 457 kPa. I can't believe the glass reaction system can work under such high inside pressure, especially the valves sealed by grease, and **(c)** how is the accuracy (and applicable range) of the calibration curve for the H₂ gas?

Response: The question is divided into three parts: (a), (b), and (c), each of which will be addressed separately.

Question (a): Activity degradation of D-O-ZIS/Pt/CoO_x in hydrogen evolution half-reaction.

Response: Actually, the high photocatalytic activity and stability of D-O-ZIS/Pt/CoO_x for the hydrogen production half-reaction is ascribed to the result of “distortion-evoked cation-site oxygen doping” strategy. This strategy generates large electronegativity differences between adjacent atomic sites, with atomic S sites being more electron-rich for favorable hydrogen adsorption. The strong charge redistribution character activates stable hydrogen adsorption/desorption at atomic S sites, inducing a high hydrogen evolution photocatalytic activity (this is demonstrated by the DFT charge calculation and free energy barrier calculation of hydrogen in **Fig. 5** and **6**, as well as in-situ Raman spectroscopy). The increase of the oxidation energy barrier of S ion in D-O-ZIS indicates that S atom has a high stability during hydrogen evolution (**Fig. 6f**). In addition, the loading of co-catalysts effectively facilitates the extraction of electrons and sacrificial reagents consume holes, further enhancing the stability over photocatalytic hydrogen evolution. The hydrogen production tests were performed multiple times, revealing minor fluctuations, and maintaining consistent stability.

Question (b): What is the volume of the empty space of the reaction system, and whether the pressure generated by the reaction gas can exist in the Labsolar 6A loop system.

Response: The total volume of empty space within the reaction system is 665 mL, comprising a 450 mL photocatalytic reactor (excluding 50 mL of reaction solution), a 150 mL gas storage bottle, and an approximate pipeline volume of 65 mL. We initially conducted a preliminary estimate of hydrogen production half-reaction in the D-O-ZIS/Pt/CoO_x using a Labsolar 6A loop system under vacuum/negative pressure conditions. The D-O-ZIS/Pt/CoO_x yielded an approximate 23 mmol hydrogen amount. Employing the ideal gas law ($PV = nRT$) with a system volume (V) of 665 mL, the calculated pressure (p) was determined to be 84 kPa corresponding to the 23 mmol of H₂, which is below the standard atmospheric pressure of 101 kPa and feasible for working in the Labsolar 6A loop system.

Question (c): How is the accuracy (and applicable range) of the calibration curve for the H₂ gas?

Response: The standard curve for hydrogen gas was constructed through multiple tests involving the injection of varying hydrogen gas volumes, consistently yielding goodness-of-fit values exceeding 0.9998. Furthermore, the fits within these three volume intervals (0-35, 35-65, 65-100 mL) closely resemble straight lines, as depicted in Fig. R1a. Hence, the quantification of H₂ production result around 0-100 mL is accurate.

Fig. R1. Photocatalytic performance. **a** The standard curve for H₂ testing of peak area obtained from the chromatographic column for each standard concentration of H₂ (Volume range: 0-35 mL, 35-65 mL, 65-100 mL). The standard curves for the volumes in these three intervals approximate a straight line; **b** Time-dependent photocatalytic hydrogen half-evolution profiles of D-O-ZIS/Pt/CoO_x using drainage method, Na₂S/Na₂SO₃ acts as holes sacrificial agent. The mass of photocatalysts is 35 mg, Pt and CoO_x as cocatalyst, under standard AM 1.5 illumination (100 mW cm⁻²). Error bars represent the standard deviations from the statistic results of three sets of experiments.

Due to the produced hydrogen amount of D-O-ZIS/Pt/CoO_x in preliminary experiments far beyond the limitation of standard curve for hydrogen gas, we employed the drainage method to elucidate its photocatalytic H₂ evolution activity (*Nat Energy* 1, 16151, 2016). Typically, photocatalytic test for drainage method was measured in an off-line reaction vessel of glass bottle (500 mL). 35 mg of the prepared photocatalyst was dispersed into 100 mL of deionized water. After ultrasonically dispersing for 30 s, the solution was bubbled with Ar for 20 min to remove any air in the reaction cell. A 300 W Xe lamp was used as the light source. As shown in Fig. R1b, the hydrogen production corresponding to the given drainage volume was then calculated using the gas molar formula. The drainage method measurements were repeated multiple times

to ensure data accuracy, and error bars were incorporated into the results. Based on the measured drainage volume of 484 mL in 12 h, the hydrogen production of D-O-ZIS/Pt/CoO_x was determined to be about 21.6 mmol.

In the revised version, **Fig. R1a** is added as **Supplementary Fig. 1e** in the supporting information; and the relative discussion has been added in the page 18 of supporting information.

2. *The amount of O₂ for the reverse reaction test is too small compared with the produced H₂ in Figure R7b. For overall water splitting reactions, the produced amount of H₂ and O₂ are comparable (2:1 in molar ratio). Therefore, it would make sense to introduce a comparable amount of O₂ for the reverse reaction examination, for example, to add around 12 mmol O₂ in Figure R7b. It is not necessary to use isotopic ¹⁸O₂; quantification of the change in the amount of H₂ and O₂ by gas chromatography is enough to address the issue. The same for Figure R9b, in which ~12 mmol H₂ should be introduced. In addition, it is incorrect to use the term “photocatalytic overall water splitting” in the captions of Figure R7 and R9 because these are sacrificial H₂ and O₂ evolution reactions (NOT overall water splitting).*

Response: Thanks very much for the valuable comments. We redesigned the experiment to investigate the reverse hydrogen-oxygen recombination reaction according Reviewer’s suggestion. Isotope experiment was removed and the terms such as “photocatalytic overall water splitting” in captions were revised.

After conducting the hydrogen or oxygen evolution half-reactions on the D-O-ZIS/Pt/CoO_x photocatalyst, stoichiometric amounts of O₂ or H₂ were introduced to investigate the hydrogen-oxygen recombination. **Fig. R2a** illustrates that D-O-ZIS/Pt/CoO_x generated O₂ gas of 597 umol after a 1-hour photocatalytic oxygen evolution half-reaction under standard AM1.5G illumination (100 mW cm⁻²). Following this, illumination was removed, and about 1194 umol of H₂ was introduced in an approximate 2:1 ratio to the generated O₂ gas, initiating a subsequent dark reaction. Under dark conditions, the amounts of H₂ and O₂ gradually decreased along with time at an approximate stoichiometric ratio of 2:1, confirming hydrogen-oxygen recombination. After 12 h dark reaction, about 165.7 umol of H₂ and 82.1 umol of O₂

gases were consumed, indicating a 13.8% decline of H₂ and O₂ gas (Fig. R2a, b). Similarly, after hydrogen evolution half-reaction on D-O-ZIS/Pt/CoO_x, 966 μmol O₂ was introduced in an approximate 1:2 ratio to the generated H₂ gas (1933 μmol) under dark reaction and the H₂ and O₂ gas exhibits a 13.3% decline (Fig. R2c, d). Thus, reverse hydrogen-oxygen recombination reaction occurs at D-O-ZIS/Pt/CoO_x surface, resulting in an approximate 13-14% decline after 12 h dark reaction.

Fig. R2. Hydrogen-oxygen recombination reactions on D-O-ZIS/Pt/CoO_x photocatalyst. a Hydrogen–oxygen recombination reaction after 1h photocatalytic oxygen evolution half-reaction, the injected H₂ amount is 1194 μmol; **b** The corresponding amounts of H₂ and O₂ consumed in the hydrogen–oxygen recombination reaction; **c** Hydrogen–oxygen recombination reaction after 1h hydrogen evolution half-reaction, the injected O₂ amount is 966 μmol; **d** The corresponding amounts of H₂ and O₂ consumed in the hydrogen–oxygen recombination reaction. Error bars represent the standard deviations from the statistic results of three sets of experiments.

In the revised version, **Fig. R2** is added as new **Supplementary Fig. 17** in the supporting information; and the relative discussion has been added in the page 44 of supporting information and copied below:

“Supplementary Fig. 17a illustrates that D-O-ZIS/Pt/CoO_x generated O₂ gas of 597 μmol after a 1-hour photocatalytic oxygen evolution half-reaction under standard AM1.5G illumination (100 mW cm⁻²). Following this, illumination was removed, and about 1194 μmol of H₂ was introduced in an approximate 2:1 ratio to the generated O₂ gas, initiating a subsequent dark reaction. Under dark

conditions, the amounts of H_2 and O_2 gradually decreased with time at an approximate stoichiometric ratio of 2:1, confirming hydrogen-oxygen recombination. After 12 h dark reaction, about 165.7 μmol of H_2 and 82.1 μmol of O_2 gases were consumed, indicating a 13.8% decline of H_2 and O_2 gas (**Supplementary Fig. 17a, b**). Similarly, after hydrogen evolution half-reaction on D-O-ZIS/Pt/CoO_x, 966 μmol O_2 was introduced in an approximate 1:2 ratio to the generated H_2 gas (1933 μmol) under dark reaction and the H_2 and O_2 gas exhibits a 13.3% decline (**Supplementary Fig. 17c, d**). Thus, reverse hydrogen-oxygen recombination reaction occurs at D-O-ZIS/Pt/CoO_x surface, resulting in an approximate 13-14% decline after 12 h dark reaction ^[27].” (in the page 44 of supporting information)

3. How about the reverse reaction over D-O-ZIS/Pt/CoO_x under dark conditions? That is, introduce a certain amount of H_2 and O_2 with a molar ratio of 2:1 (for example, 4 mmol H_2 and 2 mmol O_2) to the reaction system and check the time-dependent change in the amount of H_2 and O_2 without light irradiation. This simple experiment is what we initially suggest the authors do, for the sake of figuring out the reverse reactions over the bare Pt on the surface.

Response: According to reviewer suggestion, we introduced 372 μmol H_2 and 186 μmol O_2 at an approximate 2:1 ratio into the system under dark reaction to investigate hydrogen-oxygen recombination. During the dark conditions, the amounts of H_2 and O_2 gradually decreased, indicating the occurrence of the reverse recombination reaction between hydrogen and oxygen. After 12 h of the dark reaction, approximately 51.7 μmol of H_2 and 25.3 μmol of O_2 gases were consumed, resulting in a reduction of 13.9% (**Fig. R3a, b**).

Furthermore, we conducted photocatalytic overall water splitting, followed by dark reactions on the D-O-ZIS/Pt/CoO_x catalyst, to further study hydrogen-oxygen recombination. As depicted in **Fig. R3c, d**, nearly stoichiometric amounts of H_2 (373 μmol) and O_2 (178 μmol) were initially generated in pure water under light irradiation. Subsequently, upon removal of the light source, the quantities of H_2 and O_2 gradually decreased over time, maintaining an approximate stoichiometric ratio of 2:1. This gradual decline further illustrates the occurrence of hydrogen-oxygen recombination.

Ultimately, around 52.9 μmol of H_2 and 25.8 μmol of O_2 gas were consumed, indicating a decrease of 14.1%. Hence, combined with the O_2 and H_2 adsorption energy results (Supplementary Fig. 16), we can conclude that the generated H_2 and O_2 will recombine at Pt site of D-O-ZIS/Pt/CoO_x during photocatalytic process with the proportion of hydrogen-oxygen recombination occurring at approximately 14% after 12 h dark reaction.

Fig. R3. Hydrogen-oxygen recombination reactions on D-O-ZIS/Pt/CoO_x photocatalyst. **a** Hydrogen-oxygen recombination reaction by injecting stoichiometric amounts of H_2 and O_2 into the system, the injected H_2 amount is 372 μmol and the injected O_2 amount is 186 μmol ; **b** The corresponding amounts of H_2 and O_2 consumed in the hydrogen-oxygen recombination reaction; **c** Hydrogen-oxygen recombination reaction after photocatalytic overall water splitting; **d** The corresponding amounts of H_2 and O_2 consumed in the hydrogen-oxygen recombination reaction. Error bars represent the standard deviations from the statistic results of three sets of experiments.

In the revised version, **Fig. R3** is added as new **Supplementary Fig. 18** in the supporting information; two new references from Prof. Domen and Prof. Can Li have been cited as Ref. 38 and Ref. 39; and the relative discussion has been added in the page 8 of main text, pages 10 and 45-46 of supporting information and copied below:

“Hydrogen-oxygen recombination reaction occurred over time at Pt site of D-O-ZIS/Pt/CoO_x during photocatalytic process, along with an approximately 14% decline of H_2 and O_2 amounts after

12 h dark reaction (**Supplementary Fig. 17, 18**) [1, 38, 39].” (in the page 8 of main text)

“**Water formation reaction.** The H_2 and O_2 recombination reaction was investigated in the closed gas circulation system. The sample was placed at the bottom of the reactor, and then, H_2 and O_2 gases in a stoichiometric ratio of 2:1 was introduced or generated in the system. The reduction in gas content resulting from water formation reaction in the dark was then monitored by gas chromatography at each interval.” (in the page 10 of supporting information)

“As depicted in **Supplementary Fig. 18a, b**, we introduced a specific amount of H_2 (372 μmol) and O_2 (186 μmol) into the system at an approximate 2:1 ratio for a subsequent 12-hour dark reaction. During the dark conditions, the amounts of H_2 and O_2 gradually decreased, indicating the occurrence of the reverse recombination reaction between hydrogen and oxygen. After 12 h of reaction, approximately 51.7 μmol of H_2 and 25.3 μmol of O_2 gases were consumed, reflecting a decrease of 13.9%. As shown in **Supplementary Fig. 18c, d**, initially, nearly stoichiometric amounts of H_2 (373 μmol) and O_2 (178 μmol) in a 2:1 ratio were generated under light. Upon removing light, H_2 and O_2 quantities gradually decreased while maintaining an approximate 2:1 ratio. This gradual decline further illustrates the occurrence of hydrogen-oxygen recombination. Ultimately, around 52.9 μmol of H_2 and 25.8 μmol of O_2 gas were consumed, indicating a decrease of 14.1% after 12 h dark reaction, consistent with findings from injecting stoichiometric H_2 and O_2 amounts for recombination. Hence, combined with the O_2 and H_2 adsorption energy results (**Supplementary Fig.16**), we can conclude that the generated H_2 and O_2 will recombine at Pt site of D-O-ZIS/Pt/CoO_x during photocatalytic process with the proportion of hydrogen-oxygen recombination occurring at approximately 14% after 12 h dark reaction [27, 28].” (in the page 45-46 of supporting information)

4. The authors show the standard curve for O_2 in Figure R15a, in which 3 mL (equivalent to around 134 μmol) O_2 at maximum was used for the calibration. Considering that the response coefficient of TCD is not linearly proportional to the amount of O_2 , the applicable range of this standard curve should be limited to around 134 μmol . However, the tested O_2 amount was beyond 1200 μmol . This makes me doubt again the accuracy of the data in this work.

Response: Thanks very much for the valuable comments. We recalibrate the standard curve of O_2 and revised corresponding results. The range of new O_2 standard

curve was between 0-36 mL (**Fig. R4a**). Thirteen standard volume sample points were taken within this range. A consistent volume of standard oxygen was introduced every 30 minutes and passed through the chromatographic column for testing. The revised oxygen yield of O₂ evolution half-reaction for D-O-ZIS is determined to be 1464.7 μmol, which encompassed in the standard curve for oxygen testing (**Fig. R4b**). Meanwhile, all the oxygen yield in this paper was recalculated via the new standard curve for oxygen testing.

Fig. R4. Photocatalytic performance. **a** The standard curve for O₂ testing of peak area obtained from the chromatographic column for each standard concentration of O₂ (Volume range: 0-36 mL); **b** Time-dependent photocatalytic O₂ evolution half-reaction for ZIS, D-ZIS and D-O-ZIS in the presence of sacrificial reagents (20 mM NaIO₃ aqueous solution). Error bars represent the standard deviations from the statistic results of three sets of experiments.

In the revised version, **Fig. R4a** and **Fig. R4b** are updated as new **Supplementary Fig. 1f** and **Supplementary Fig. 21c** in the supporting information; and the oxygen production in **Fig. 3a, d**, **Supplementary Fig. 9a-c**, **12b**, **19a**, **26**, **Supplementary Tables 4**, **8**, **12** have been examined according to the new standard curve for oxygen testing in the main text and supporting information. The relative discussion has been added in the page 18 of supporting information.

5. The comment of “why was the activity almost the same by loading significantly different ratios of Pt/CoO_x (1:4 and 3:1) in Figure S7a and b” is not well addressed. Is that a coincidence or is there any underlying reason?

Response: The closed photocatalytic activity for Pt:CoO_x ratios of 1:4 and 3:1 is coincidental. At a Pt to CoO_x content loading ratio of 1:4, the D-O-ZIS/Pt/CoO_x exhibits the highest photocatalytic H₂ and O₂ evolution rate with a value of 31.1 and 14.8 μmol h⁻¹, respectively. It is evident that increasing amount of CoO_x loading

initially improves performance, providing more active sites and promoting charge separation and transfer. However, excessive CoO_x loading amount hampers catalytic sites, causing a shielding effect and performance decline (*Chem. Soc. Rev.*, 43, 7787, 2014). Additionally, increased Pt loading enhances photocatalytic water splitting. At Pt to CoO_x ratio 3:1, D-O-ZIS/Pt/ CoO_x exhibits a slight decrease, yielding 25.55 and 11.87 $\mu\text{mol h}^{-1}$ for H_2 and O_2 evolution, respectively.

6. *The deposition method of CoO_x in this work is very strange. What is the particle size and morphology of the as-prepared CoO_x particles? I do not believe that just mixing the CoO_x powder and D-O-ZIS in water would form CoO_x -decorated D-O-ZIS. In my opinion, this step will form segregated CoO_x and D-O-ZIS, without intimate contact between them. A widely used method for unselectively depositing the CoO_x cocatalyst is impregnation with $\text{Co}(\text{NO}_3)_2$ (or other cobalt precursors) followed by thermal treatment. Did the authors try the impregnation method?*

7. *HRTEM is not reliable to analyze the CoO_x and Pt species on D-O-ZIS. The HRTEM image (Figure S8a) is unduly treated, making it hard to recognize the real surface structure (and the original HRTEM image). STEM-EDS should be performed to clearly understand whether or not CoO_x was deposited (see the comment above) and the nanostructures of Pt (and CoO_x) formed.*

Response: Thanks very much for the valuable comments. Questions 6 and 7 can be addressed together.

Sorry, we apologize for the mistake in describing the CoO_x cocatalyst deposition method in the original version. Actually, the CoO_x cocatalyst was loaded via an impregnation-photo-deposition method (*Nature*, 613, 66, 2023; *Angew. Chem. Int. Ed.*, e202304694, 2023). According to the reviewer's comment, in the revised version, we also employed the impregnation-thermal-treatment method to load CoO_x . We investigated the morphology and photocatalytic activity of D-O-ZIS/Pt/ CoO_x prepared via the two methods and discussed below.

As for the **impregnation-photo-deposition method** for CoO_x loading, 35 mg D-O-ZIS photocatalyst, H_2PtCl_6 (0.5 mg) and $\text{Co}(\text{NO}_3)_2 \cdot 6\text{H}_2\text{O}$ (2.1 mg) was injected into the chamber with 50 mL deionized water and the chamber was irradiated under a 300-W Xe lamp (AM1.5G, 100 mW cm^{-2}) for 30 min. After the light irradiation, the obtained photocatalyst was centrifuged and washed by deionized water and then dried at 80°C , yielding D-O-ZIS/Pt/ CoO_x catalyst. As shown in **Fig. R5a, b**, the CoO_x nanoparticles loading on the D-O-ZIS surface with size around 5 nm. The co-existence of (311) plane of CoO_x and (102) plane of D-O-ZIS further proves the CoO_x nanoparticles successful loading on the surface of D-O-ZIS. Meanwhile, the Pt nanoparticles with diameters of approximately 5-6 nm and 0.22 nm lattice distances corresponding to (111) plane also observed. The energy-dispersive X-ray spectroscopy (EDX) elemental mapping results in **Fig. R5c** further prove the intimate contact between Pt, CoO_x and D-O-ZIS.

Fig. R5. Morphology and elemental distribution of D-O-ZIS modified Pt and CoO_x photocatalyst. **a** TEM image of D-O-ZIS/Pt/ CoO_x at low magnification; **b** HRTEM image of D-O-ZIS/Pt/ CoO_x ; **c** Scanning TEM image (Embedded in **Fig. R5a**) and the corresponding EDX elemental mapping images of the distribution of Zn, In, S, O, Co, and Pt species on D-O-

ZIS/Pt/CoO_x. The scale (20 nm) applies to the images in **Fig. R5c**.

As for the **impregnation-thermal-treatment method** for CoO_x loading, Co(NO₃)₂·6H₂O (2.1 mg) and D-O-ZIS (35 mg) were dispersed in 50 mL of deionized water under continuous stirring. The solution was heated on a heating plate until complete water evaporation, and the resulting precipitate was then subjected to thermal decomposition in a muffle furnace at 400 °C for 1 hour. Following the thermal treatment, D-O-ZIS/CoO_x was obtained. Afterward, the obtained D-O-ZIS/CoO_x photocatalyst and H₂PtCl₆ (333 μL, 1.5 mg mL⁻¹) were dispersed in an aqueous solution for Pt loading. Following light irradiation under standard AM1.5G illumination (100 mW cm⁻²) for 30 min, the photocatalyst was centrifuged, washed with deionized water, and then dried, resulting in the D-O-ZIS/Pt/CoO_x catalyst. The D-O-ZIS/Pt/CoO_x catalyst prepared via the impregnation-heat treatment method exhibited relatively larger particles. The TEM image reveals Pt nanoparticles measuring 6-8 nm and CoO_x nanoparticles of approximately 7-12 nm, attached to the D-O-ZIS surface (**Fig. R6a, b**). The HRTEM image showed lattice fringes at 0.22 nm for the (111) plane of Pt nanocrystalline and 0.25 nm for the (311) plane of CoO_x. The EDX elemental mapping results in **Fig. R6c** confirm the presence of distributed CoO_x and Pt nanoparticles on the surface of D-O-ZIS.

Furthermore, the photocatalytic performance of the resultant D-O-ZIS/Pt/CoO_x photocatalyst, loaded with cocatalysts through the impregnation-heat treatment method, was also evaluated. As shown in **Fig. R6d**, the D-O-ZIS/Pt/CoO_x exhibits photocatalytic H₂ and O₂ evolution rate of 23.2 and 11.4 μmol h⁻¹, respectively. The photocatalytic performance of D-O-ZIS/Pt/CoO_x, loaded with cocatalysts via the impregnation-heat treatment method, demonstrated a decreased photocatalytic activity compared with that prepared by impregnation-photo-deposition method (photocatalytic H₂ and O₂ evolution rate of 31.1 and 14.8 μmol h⁻¹).

Fig. R6. Morphology and photocatalytic performance of D-O-ZIS modified Pt and CoO_x photocatalyst by utilizing the impregnation-heat treatment method. **a** TEM image of D-O-ZIS/Pt/ CoO_x at low magnification; **b** HRTEM image of D-O-ZIS/Pt/ CoO_x ; **c** Scanning TEM image (Embedded in Fig. R6a) and the corresponding EDX elemental mapping images of the distribution of Zn, In, S, O, Co, and Pt species on D-O-ZIS/Pt/ CoO_x . The scale (20 nm) applies to the images in Fig. R6c; **d** Time-dependent photocatalytic overall water splitting over D-O-ZIS/Pt/ CoO_x in pure water under standard AM 1.5 illumination (100 mW cm^{-2}). $1.5 \text{ mg mL}^{-1} \text{ H}_2\text{PtCl}_6$, $2.1 \text{ mg Co(NO}_3)_2 \cdot 6\text{H}_2\text{O}$, the photocatalyst mass was 35 mg and the photocatalytic activity was evaluated via the total hydrogen and oxygen yield of a cycle, the time of each cycle is 12 h. Error bars represent the standard deviations from the statistic results of three sets of experiments.

In the revised version, Fig. R5 and Fig. R6 are added as a new Supplementary Fig. 8 and Supplementary Fig. 10 in the supporting information; and the relative discussion has been added in the page 6, 27-28 and 31-32 of supporting information and copied below:

“Impregnation-photo-deposition method loaded CoO_x cocatalyst on D-O-ZIS. Typically,

35 mg D-O-ZIS photocatalyst, H_2PtCl_6 (0.5 mg) and $Co(NO_3)_2 \cdot 6H_2O$ (2.1 mg) was injected into the chamber with 50 mL deionized water and the chamber was irradiated under a 300-W Xe lamp (AM1.5G, 100 mW cm^{-2}) for 30 min. After the light irradiation, the obtained photocatalyst was centrifuged and washed by deionized water and then dried at 80 °C, yielding D-O-ZIS/Pt/ CoO_x catalyst.” (in the page 6 of supporting information)

“As shown in **Supplementary Fig. 8**, TEM image of the D-O-ZIS/Pt/ CoO_x photocatalyst by the impregnation-photo-deposition method depicts a uniform distribution of Pt nanoparticles and CoO_x nanoparticles on the D-O-ZIS surface (**Supplementary Fig. 8a**). To further elucidate the interfacial contact between CoO_x and D-O-ZIS, we measured the high-resolution transmission electron microscopy (HRTEM) image of D-O-ZIS/Pt/ CoO_x (**Supplementary Fig. 8b**). The Pt nanoparticles exhibited diameters of approximately 5-6 nm, and the CoO_x nanoparticles measured around 5 nm on the surface of D-O-ZIS. A small number of large particles appeared in the morphology. The resolved lattice distances of 0.22 nm match well with the (111) plane of Pt, while a lattice distance of 0.25 nm is attributed to the (311) plane of CoO_x . Both Pt and CoO_x are closely connected with the (102) plane of D-O-ZIS measuring 0.32 nm. The energy-dispersive X-ray spectroscopy (EDX) elemental mapping results in **Supplementary Fig. 8c** further prove the intimate contact between Pt, CoO_x and D-O-ZIS^[25-30].” (in the page 27-28 of supporting information)

“We also used the impregnation-heat treatment method to load Pt and CoO_x nanoparticles onto D-O-ZIS. In detail, $Co(NO_3)_2 \cdot 6H_2O$ (2.1 mg) and D-O-ZIS (35 mg) were dispersed in 50 mL of deionized water under continuous stirring. The solution was heated on a heating plate until complete water evaporation, and the resulting precipitate was then subjected to thermal decomposition in a muffle furnace at 400 °C for 1 hour. Following the thermal treatment, D-O-ZIS/ CoO_x was obtained. Afterward, the obtained D-O-ZIS/ CoO_x photocatalyst and H_2PtCl_6 ($333\ \mu\text{L}$, 1.5 mg mL^{-1}) were dispersed in an aqueous solution for Pt loading. Following light irradiation under standard AM1.5G illumination (100 mW cm^{-2}) for 30 min, the photocatalyst was centrifuged, washed with deionized water, and then dried, resulting in the D-O-ZIS/Pt/ CoO_x catalyst. The D-O-ZIS/Pt/ CoO_x catalyst prepared using the impregnation-heat treatment method exhibited relatively large particles. **Supplementary Fig. 10a, b** displayed Pt

particles around 6-8 nm and CoO_x nanoparticles about 7-12 nm in size with aggregation, attached to the D-O-ZIS surface. HRTEM revealed a lattice fringe distance of 0.22 nm corresponding to the (111) plane of Pt nanocrystals, and 0.25 nm for the (311) plane of CoO_x . The EDX elemental mapping results in **Supplementary Fig. 10c** confirm the presence of distributed CoO_x and Pt nanoparticles on the surface of D-O-ZIS.

Furthermore, the photocatalytic performance of the resultant D-O-ZIS/Pt/ CoO_x photocatalyst, loaded with cocatalysts through the impregnation-heat treatment method, was also evaluated. As shown in **Supplementary Fig. 10d**, the D-O-ZIS/Pt/ CoO_x exhibits photocatalytic H_2 and O_2 evolution rate of 23.2 and 11.4 $\mu\text{mol h}^{-1}$, respectively. The photocatalytic performance of D-O-ZIS/Pt/ CoO_x , loaded with cocatalysts via the impregnation-heat treatment method, demonstrated a decreased photocatalytic activity compared with that prepared by impregnation-photo-deposition method. (photocatalytic H_2 and O_2 evolution rate of 31.1 and 14.8 $\mu\text{mol h}^{-1}$)” (in the page 31-32 of supporting information)

8. EPR is a technique for detecting unpaired electrons. A vacancy itself will not produce an EPR signal. When (unpaired) free electrons were trapped in specific vacancies, it may generate an EPR signal at g value of around 2.0023. Therefore, the expressions of “...g = 2.004 attributed to the Zn vacancy” and similar were not scientifically correct. According to the authors’ response, I do not think that they understand EPR and this issue was not addressed in the revised manuscript.

Response: We revised the ESR description to specify unpaired electron capture by zinc vacancies. “Electron spin resonance (ESR) spectroscopy of D-ZIS displayed a peak intensity at g = 2.004 attributed to the unpaired free electrons trapped in Zn vacancy, confirming the existence of Zn vacancy (**Fig. 2b**).” The description has been updated in revised main text on page 6.

Reviewers' comments:

Reviewer #2 (Remarks to the Author):

In the response of question 1(b), the authors claimed that the volume of the reactor was 450 mL. However, they indicated that the reaction volume was 100 mL in the previous version of SI (Line 9-10, Page 10, File name: 429087_1_supp_7837857_rxq676). In the current version of SI, the authors have intentionally deleted the statement of reactor volume of "100 mL" (Line 1, Page 10, File name: 429087_2_supp_8062363_s0xjc0) without any explanation in the rebuttal letter. If it were true, the experimental design would be obviously unreasonable because a 50 mL solution was used in a 450 mL reactor (it should reasonably be a 50 mL solution in a 100 mL reactor as described in the previous version of SI). Another suspicious point is that the authors newly claimed that they were actually using the photodeposition method for the CoOx deposition, which was completely different from the previous version in which the physical mixing method was used. These serious issues make me highly suspect the reliability of the data, and I would reject this work steadfastly. Other concerns are provided as follows.

1. CoOx was thermodynamically very hard to be photodeposited onto the surface of narrow-band-gap semiconducting materials. High-resolution EDS analysis of the marked particles in Fig. R5b should be provided.
2. It is very strange that when CoOx was deposited by impregnation-thermal-treatment method in air at 673 K for 1 h, the resulting sample could evolve H₂ and O₂ with relatively high rates (Fig. R6). I do not think that ZnIn₂S₄ would be stable under such a condition; this treatment would definitely destroy the OWS performance.
3. It is surprising to see that the water formation reaction was minor over bare Pt-decorated (without an overlay such as the typical Cr₂O₃) photocatalyst. The authors should explain the detailed reason.

Point-by-point Response to the Reviewers' Comments (Round 3)

NOTE: Review #1 and #3 are satisfied with our revision and recommend publication in the present form in Round 2 revision.

Reviewer #1 (Remarks to the Author):

Thanks to Li and Xu et al. for their thoughtful responses to my previous questions. Overall, the answers were quite rational and satisfying. Therefore, I suggest its publication on Nat Commun.

Response: Thanks for your comment and recognition.

Reviewer #3 (Remarks to the Author):

The authors have addressed all my comments. Therefore, I'd like to recommend its publication in the present form now.

Response: Thanks for your comment and recognition.

Reviewer #2:

In the response of question 1(b), the authors claimed that the volume of the reactor was 450 mL. However, they indicated that the reaction volume was 100 mL in the previous version of SI (Line 9-10, Page 10, File name: 429087_1_supp_7837857_rxq676). In the current version of SI, the authors have intentionally deleted the statement of reactor volume of "100 mL" (Line 1, Page 10, File name: 429087_2_supp_8062363_s0xjc0) without any explanation in the rebuttal letter. If it were true, the experimental design would be obviously unreasonable because a 50 mL solution was used in a 450 mL reactor (it should reasonably be a 50 mL solution in a 100 mL reactor as described in the previous version of SI).

Response: The reaction vessel used for photocatalytic gas evolution consisted of two types: 100 mL and 500 mL. The 100 mL reactor was employed for photocatalytic overall water splitting tests due to the lower gas production. Meanwhile, the 500 mL reactor was utilized in the preliminary tests for the hydrogen production half-reaction.

Since the quantity of generated hydrogen surpassed the range covered by the standard hydrogen gas curve, we subsequently adopted a drainage method to precisely quantify the yield. Thus, in the revised version, we removed the reactor vessel volume information from the Supplementary Information.

Another suspicious point is that the authors newly claimed that they were actually using the photodeposition method for the CoO_x deposition, which was completely different from the previous version in which the physical mixing method was used. These serious issues make me highly suspect the reliability of the data, and I would reject this work steadfastly. Other concerns are provided as follows.

Response: In the initial version, there was an error in the catalyst loading method due to oversight. This mistake has been corrected. Meanwhile, we also employed two methods for the CoO_x deposition to respond the reviewer's comment. The correction will not affect the main conclusion and novelty of our work.

1. CoO_x was thermodynamically very hard to be photodeposited onto the surface of narrow-band-gap semiconducting materials. High-resolution EDS analysis of the marked particles in Fig. R5b should be provided.

Response: The photodeposition method for loading Pt and CoO_x is a commonly used and straightforward approach for depositing cocatalysts. It is noteworthy that the selectivity of CoO_x loading on semiconductors is not particularly high, especially when defects and vacancies are present in the material. Research literature has reported the loading CoO_x on both narrow-bandgap and wide-bandgap semiconductor surfaces, as exemplified by studies on InGaN/GaN/CoO_x (*Nature*, **613**, **66**, **2023**); Poly (triazine imide) (PTI)/CoO_x/Rh-Cr₂O₃ (*Angew. Chem. Int. Ed.*, **e202304694**, **2023**); BiVO₄/CoO_x (*Nat. Commun.*, **13**, **484**, **2022**); ZrO₂/TaON/BiVO₄-CoO_x (*Joule*, **2**, **2393**, **2018**); Ovacancy-TiO₂-CoO_x (*Chem. Eng. J.*, **439**, **135744**, **2022**); poly (Triazine Imide)/CoO_x (*Angew. Chem. Int. Ed.*, **61**, **e20211338**, **2022**); MoS₂/CoO_x (*J. Mater. Sci. Technol.*, **124**, **171**, **2022**); ZnS/CdS/CoO_x (*Sci. Bull.*, **63**, **2018**); g-C₃N₄/CoO_x (*Angew. Chem. Int. Ed.*, **62**, **e2023046**, **2023**); BiOBr/CoO_x/g-C₃N₄ (*J. Alloys Compd.*, **875**, **2021**); BiVO₄/CoO_x (*J. Colloid Interf. Sci.*, **646**, **2023**); and Bi₂MoO₆/CoO_x

(*Chem. Eng. J.*, 325, 1, 690).

Actually, the co-existence of CoO_x (311) plane and ZnIn_2S_4 (102) plane and energy-dispersive X-ray spectroscopy (EDX) elemental mapping have already been provided (**Supplementary Fig. 10 b, c, i.e., Fig. R5** in the response letter), which provide enough evidence to successful loading CoO_x on photocatalyst surface.

2. *It is very strange that when CoO_x was deposited by impregnation-thermal-treatment method in air at 673 K for 1 h, the resulting sample could evolve H_2 and O_2 with relatively high rates (Fig. R6). I do not think that ZnIn_2S_4 would be stable under such a condition; this treatment would definitely destroy the OWS performance.*

Response: The prepared D-O-ZIS is a thermally stable material, which undergo thermal treatment at 500 °C during synthesis process (**Fig. 1a** in the main text). The D-O-ZIS/ $\text{Co}(\text{NO}_3)_2$ precursor underwent annealing at 400 °C in a muffle furnace for 1 hour, which will not significantly affect the photocatalytic activity. In addition, the photocatalytic performance of D-O-ZIS/Pt/ CoO_x , loaded with cocatalysts via the impregnation-heat treatment method, demonstrated a decreased photocatalytic activity compared with that prepared by impregnation-photo-deposition method. The impregnation-photo-deposition method is mainly used in our work.

3. *It is surprising to see that the water formation reaction was minor over bare Pt-decorated (without an overlay such as the typical Cr_2O_3) photocatalyst. The authors should explain the detailed reason.*

Response: The D-O-ZIS/Pt/ CoO_x exhibits a moderate reverse reaction of water formation during the photocatalytic overall water splitting with a value of approximately 14%, which is greater than typical photocatalyst loading Pt (approximately 29%, *J. Am. Chem. Soc.* 143, 10633, 2021) and lower than the photocatalyst overlay Cr_2O_3 (6% *Angew. Chem. Int. Ed.*, e202304694, 2023).

The reason for the suppressed water formation reaction of D-O-ZIS/Pt/ CoO_x can be ascribed to the optimized adsorption energies. As shown in **Supplementary Information Figure 11**, we have calculated the adsorption energies of the intermediate species H and OH, as well as the reverse reaction species H_2 and O_2 , involved in the

redox reactions occurring on the surface of D-O-ZIS/Pt/CoO_x. By comparing the adsorption energies of these intermediate species, Pt-H exhibits a more negative adsorption energy of -0.61 eV, which is lower than the adsorption energies of Pt-H₂ (-0.29 eV) and Pt-O₂ (-0.23 eV). This indicates that the D-O-ZIS/Pt/CoO_x exhibits stronger adsorption of active hydrogen, making it less likely for hydrogen gas to adsorb on the catalyst surface, thereby reducing the sites available for the reverse reaction. Furthermore, CoO_x exhibits a more negative adsorption energy of -0.36 eV for OH, indicating that the D-O-ZIS/Pt/CoO_x surface is more favourable for the generation of O₂. The lower adsorption energies of H₂ and O₂ suggest that these species are less likely to adsorb on the surface and undergo subsequent reverse reactions.